# LLaMA-Omni: Seamless Speech Interaction with Large Language Models

**Qingkai Fang**[1,3]**, Shoutao Guo**[1,3]**, Yan Zhou**[1,3]**, Zhengrui Ma**[1,3]**,**
**Shaolei Zhang**[1,3]**, Yang Feng**[1,2,3*]

[1] Key Laboratory of Intelligent Information Processing
  Institute of Computing Technology, Chinese Academy of Sciences (ICT/CAS)
[2] Key Laboratory of AI Safety, Chinese Academy of Sciences
[3] University of Chinese Academy of Sciences, Beijing, China
{fangqingkai21b, fengyang}@ict.ac.cn

## Abstract

Models like GPT-4o enable real-time interaction with large language models (LLMs) through speech, significantly enhancing user experience compared to traditional text-based interaction. However, there is still a lack of exploration on how to build speech interaction models based on open-source LLMs. To address this, we propose LLaMA-Omni, a novel end-to-end model architecture designed for low-latency and high-quality speech interaction with LLMs. LLaMA-Omni integrates a pretrained speech encoder, a speech adaptor, an LLM, and a streaming speech decoder. It eliminates the need for speech transcription, and can simultaneously generate text and speech responses directly from speech instructions with extremely low latency. We build our model based on the latest Llama-3.1-8B-Instruct model. To align the model with speech interaction scenarios, we construct a dataset named InstructS2S-200K, which includes 200K speech instructions and corresponding speech responses, with a style that better matches the characteristics of speech interaction scenarios. Experimental results show that compared to previous speech-language models, LLaMA-Omni provides better responses in both content and style, with a response latency as low as 236ms. Additionally, training LLaMA-Omni takes less than 3 days on just 4 GPUs, paving the way for the efficient development of speech-language models in the future.[1]

## 1 Introduction

Large language models (LLMs), represented by ChatGPT (OpenAI, 2022), have become powerful general-purpose task solvers, capable of assisting people in daily life through conversational interactions. However, most LLMs currently only support text-based interactions, which limits their application in scenarios where text input and output are not ideal. Recently, the emergence of GPT-4o (OpenAI, 2024) has made it possible to interact with LLMs through speech, responding to user's instruction with extremely low latency and significantly enhancing the user experience. However, there is still a lack of exploration in the open-source community on building such speech interaction models based on LLMs. Therefore, how to achieve low-latency and high-quality speech interaction with LLMs is a pressing challenge that needs to be addressed.

The simplest way to enable speech interaction with LLMs is through a cascaded system based on automatic speech recognition (ASR) and text-to-speech (TTS) models, where the ASR model transcribes the user's speech instruction into text, and the TTS model synthesizes the LLM's response into speech. However, since the cascaded system sequentially outputs the transcribed text, text response, and speech response, the overall system tends to have higher latency. In contrast, some multimodal speech-language models have been proposed (Zhang et al., 2023; Rubenstein et al.,

---

*Corresponding Author: Yang Feng.

[1]Code: https://github.com/ictnlp/LLaMA-Omni
  Model: https://huggingface.co/ICTNLP/Llama-3.1-8B-Omni
  Audio Samples: https://ictnlp.github.io/llama-omni-demo/

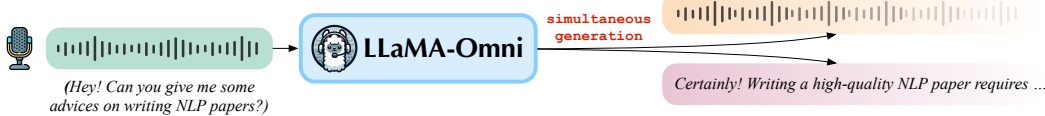

Figure 1: LLaMA-Omni can simultaneously generate text and speech responses based on the speech instruction, with extremely low response latency.

2023), which discretize speech into tokens and extend the LLM's vocabulary to support speech input and output. Such speech-language models theoretically can generate speech responses directly from speech instructions without producing intermediate text, thereby achieving extremely low response latency. However, in practice, direct speech-to-speech generation can be challenging due to the complex mapping involved, so it is common to generate intermediate text to achieve higher generation quality (Zhang et al., 2023), although this sacrifices some response latency.

In this paper, we propose a novel model architecture, LLaMA-Omni, which enables low-latency and high-quality interaction with LLMs. LLaMA-Omni consists of a speech encoder, a speech adaptor, an LLM, and a streaming speech decoder. The user's speech instruction is encoded by the speech encoder followed by the speech adaptor, and then input into the LLM. The LLM decodes the text response directly from the speech instruction, without first transcribing the speech into text. The speech decoder is a non-autoregressive (NAR) streaming Transformer (Ma et al., 2023), which takes the output hidden states of the LLM as input and uses connectionist temporal classification (CTC; Graves et al., 2006a) to predict the sequence of discrete units corresponding to the speech response. During inference, as the LLM autoregressively generates the text response, the speech decoder simultaneously generates the corresponding discrete units. To better align with the characteristics of speech interaction scenarios, we construct a dataset named InstructS2S-200K by rewriting existing text instruction data and performing speech synthesis. Experimental results show that LLaMA-Omni can simultaneously generate high-quality text and speech responses with a latency as low as 236ms. Additionally, compared to previous speech-language models like SpeechGPT (Zhang et al., 2023), LLaMA-Omni significantly reduces the required training data and computational resources, enabling the efficient development of powerful speech interaction models based on the latest LLMs.

## 2 MODEL: LLAMA-OMNI

In this section, we introduce the model architecture of LLaMA-Omni. As shown in Figure 2, it consists of a speech encoder, a speech adaptor, an LLM, and a speech decoder. We denote the user's speech instruction, text response, and speech response as $X^S$, $Y^T$, and $Y^S$ respectively.

### 2.1 SPEECH ENCODER

We use the encoder of Whisper-large-v3[2] (Radford et al., 2023) as the speech encoder $\mathcal{E}$. Whisper is a general-purpose speech recognition model trained on a large amount of audio data, and its encoder is capable of extracting meaningful representations from speech. Specifically, for the user's speech instruction $X^S$, the encoded speech representation is given by $\mathbf{H} = \mathcal{E}(X^S)$, where $\mathbf{H} = [\mathbf{h}_1, ..., \mathbf{h}_N]$ is the speech representation sequence of length $N$. We keep the speech encoder's parameters frozen throughout the entire training process.

### 2.2 SPEECH ADAPTOR

To enable the LLM to comprehend the input speech, we incorporate a trainable speech adaptor $\mathcal{A}$ that maps the speech representations into the embedding space of the LLM. Following Ma et al. (2024b), our speech adaptor first downsamples the speech representations $\mathbf{H}$ to reduce the sequence length. Specifically, every $k$ consecutive frames are concatenated along the feature dimension:

$$\mathbf{H}' = \left[\mathbf{h}'_1, ..., \mathbf{h}'_{\lfloor N/k \rfloor}\right], \text{where } \mathbf{h}'_i = \left[\mathbf{h}_{k \times (i-1)+1} \oplus \mathbf{h}_{k \times (i-1)+2} \oplus \cdots \oplus \mathbf{h}_{k \times i}\right]. \tag{1}$$

---

[2]https://huggingface.co/openai/whisper-large-v3

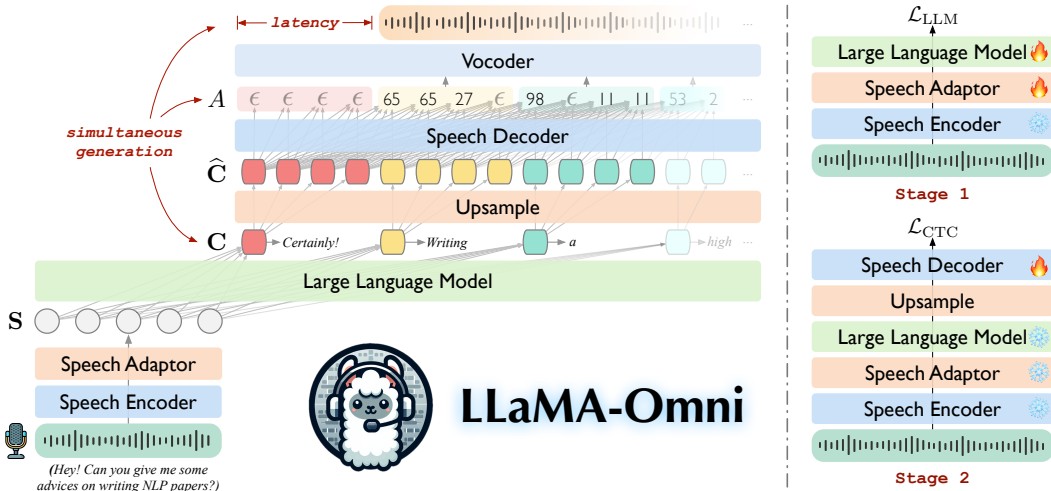

Figure 2: **Left:** Model architecture of LLaMA-Omni. **Right:** Illustration of the two-stage training strategy for LLaMA-Omni.

Next, $\mathbf{H}'$ is passed through a 2-layer perceptron with ReLU activation between the linear layers, resulting in the final speech representation $\mathbf{S}$. The above process can be formalized as follows:

$$\mathbf{S} = \mathcal{A}(\mathbf{H}) = \text{Linear}(\text{ReLU}(\text{Linear}(\text{DownSample}(\mathbf{H})))). \tag{2}$$

## 2.3 LARGE LANGUAGE MODEL

We use Llama-3.1-8B-Instruct[3] (Dubey et al., 2024) as the LLM $\mathcal{M}$, which is currently the state-of-the-art open-source LLM. It has strong reasoning capabilities and is well-aligned with human preferences. The prompt template $\mathcal{P}(\cdot)$ is shown in Appendix A. The speech representation sequence $\mathbf{S}$ is filled into the position corresponding to `<speech>`, and then the entire sequence $\mathcal{P}(\mathbf{S})$ is input into the LLM. Finally, the LLM autoregressively generates the text response $Y^T = [y_1^T, ..., y_M^T]$ directly based on the speech instruction and is trained using cross-entropy loss:

$$\mathcal{L}_{\text{LLM}} = -\sum_{i=1}^{M} \log P(y_i^T | \mathcal{P}(\mathbf{S}), Y_{<i}^T). \tag{3}$$

## 2.4 SPEECH DECODER

For the speech response $Y^S$, we first follow Zhang et al. (2023) to discretize the speech into discrete units. Specifically, we use the pretrained HuBERT (Hsu et al., 2021) model to extract continuous representations of the speech, and then convert these representations into discrete cluster indices using a K-means model. Subsequently, consecutive identical indices are merged into a single unit, resulting in the final discrete unit sequence $Y^U = [y_1^U, ..., y_L^U], y_i^U \in \{0, 1, ..., K-1\}, \forall 1 \leq i \leq L$, where $K$ is the number of clusters, and $L$ is the length of discrete unit sequence. To synthesize waveforms based on discrete units, we adopt a unit-based HiFi-GAN vocoder with a duration predictor (Polyak et al., 2021). It first predicts the duration of each discrete unit and repeats them to model prosody, and then generates the waveform based on discrete units.

To generate speech responses simultaneously with text responses, we add a streaming speech decoder $\mathcal{D}$ after the LLM. It consists of several standard Transformer (Vaswani et al., 2017) layers with the same architecture as LLaMA (Dubey et al., 2024), each containing a causal self-attention module and a feed-forward network. Similar to Ma et al. (2024a); Zhang et al. (2024b), the speech decoder runs in a non-autoregressive manner, which takes the output hidden states from the LLM as input, and generates the discrete unit sequence corresponding to the speech response. Specifically,

---

[3]https://huggingface.co/meta-llama/Meta-Llama-3.1-8B-Instruct

the output hidden states corresponding to the text response are denoted as $\mathbf{C} = [\mathbf{c}_1, ..., \mathbf{c}_M]$, where $\mathbf{c}_i = \mathcal{M}(\mathcal{P}(\mathbf{S}), Y_{<i}^T)$. We first upsample each hidden state into a chunk by a factor of $\lambda$, resulting in an upsampled hidden state sequence $\widehat{\mathbf{C}} = [\widehat{\mathbf{c}}_1, ... \widehat{\mathbf{c}}_{\lambda \cdot M}]$, where $\widehat{\mathbf{c}}_i = \mathbf{c}_{\lfloor i/\lambda \rfloor}$. Next, $\widehat{\mathbf{C}}$ is fed into the speech decoder $\mathcal{D}$, and the output hidden state sequence is denoted as $\mathbf{O} = [\mathbf{o}_1, ..., \mathbf{o}_{\lambda \cdot M}]$. We use connectionist temporal classification (CTC; Graves et al., 2006a) to align $\mathbf{O}$ with the discrete unit sequence $Y^U$. Specifically, CTC extends the output space with a special blank token $\epsilon$:

$$P(a_i|\mathbf{O}) = \text{softmax}(\mathbf{W}\mathbf{o}_i + \mathbf{b})[a_i], \forall a_i \in \{0, 1, ..., K-1, \epsilon\}, \tag{4}$$

where $\mathbf{W} \in \mathbb{R}^{(K+1) \times d}$ and $\mathbf{b} \in \mathbb{R}^{K+1}$ are weights and biases of the linear layer, and the sequence $A = [a_1, ..., a_{\lambda \cdot M}]$ is known as the *alignment*. To model the variable-length mapping between input and output, CTC introduces a collapsing function $\beta(A)$, which first merges all consecutive repeated tokens in $A$ and then eliminates all blank tokens $\epsilon$. For instance: $\beta([1, 1, 2, \epsilon, \epsilon, 2, 3]) = [1, 2, 2, 3]$. During training, CTC performs marginalization over all possible alignments as follows:

$$\mathcal{L}_{\text{CTC}} = -\log P(Y^U|\mathbf{O}) = -\log \sum_{A \in \beta^{-1}(Y^U)} P(A|\mathbf{O}) = -\log \sum_{A \in \beta^{-1}(Y^U)} \prod_{i=1}^{\lambda \cdot M} P(a_i|\mathbf{O}), \tag{5}$$

where $\beta^{-1}(Y^U)$ denotes all possible alignments of length $\lambda \cdot M$ that can be collapsed to $Y^U$. The alignment is modeled in a non-autoregressive way. During inference, we select the best alignment $A^* = \arg\max_A P(A|\mathbf{O})$, and apply the collapsing function to obtain the discrete unit sequence $\beta(A^*)$, which is then fed into the vocoder to synthesize waveform.

## 2.5 TRAINING

As shown in Figure 2, we adopt a two-stage training strategy for LLaMA-Omni. In the first stage, we train the model to generate text responses directly from the speech instructions. Specifically, the speech encoder is frozen, and the speech adaptor and the LLM are trained using the objective $\mathcal{L}_{\text{LLM}}$ in Eq. (3). The speech decoder is not involved in training during this stage. In the second stage, we train the model to generate speech responses. During this stage, the speech encoder, speech adaptor, and LLM are all frozen, and only the speech decoder is trained using the objective $\mathcal{L}_{\text{CTC}}$ in Eq. (5).

## 2.6 INFERENCE

During inference, the LLM autoregressively generates the text response based on the speech instruction. Meanwhile, since our speech decoder uses causal attention, once the LLM generates a text response prefix $Y_{\leq i}^T$, the corresponding upsampled hidden states $\widehat{\mathbf{C}}_{\leq \lambda \cdot i}$ can be fed into the speech decoder to generate a partial alignment $A_{\leq \lambda \cdot i}$, which in turn yields the discrete units corresponding to the generated text prefix. To further enable streaming synthesis of speech waveforms, when the number of generated units reaches a pre-defined chunk size $\Omega$, we input this unit segment into the vocoder to synthesize a speech segment, which is then immediately played to the user. As a result, users can start listening to the speech response without

---

**Algorithm 1:** Inference Process

**Input:** speech instruction $X^S$.
**Output:** text outputs $Y^T$, units outputs $Y^U$, waveform outputs $Y^S$.
**Model:** speech encoder $\mathcal{E}$, speech adaptor $\mathcal{A}$, LLM $\mathcal{M}$, speech decoder $\mathcal{D}$, vocoder $\mathcal{V}$.
**Require:** Minimum chunk size for units $\Omega$.
**Initialization:** $i = 1, j = 0, Y^T = []$, $\quad Y^U = [], Y^S = [], \widehat{\mathbf{C}} = []$.
$\mathbf{S} \leftarrow \mathcal{A}(\mathcal{E}(X^S));$
**while** $y_{i-1}^T \neq \langle \text{EOS} \rangle$ **do**
$\quad \mathbf{c}_i \leftarrow \mathcal{M}(\mathcal{P}(\mathbf{S}), Y_{<i}^T);$
$\quad y_i^T \leftarrow \arg\max_{y_i^T} P(y_i^T|\mathcal{P}(\mathbf{S}), Y_{<i}^T);$
$\quad Y^T \leftarrow Y^T + y_i^T;$
$\quad \widehat{\mathbf{C}} \leftarrow \widehat{\mathbf{C}} + \text{UpSample}(\mathbf{c}_i);$
$\quad \mathbf{O} \leftarrow \mathcal{D}(\widehat{\mathbf{C}});$
$\quad A^* \leftarrow \arg\max_A P(A|\mathbf{O});$
$\quad Y^U \leftarrow \beta(A^*);$
$\quad$ **if** $|Y^U| - j \geq \Omega$ **then**
$\quad\quad y^S \leftarrow \mathcal{V}(Y_{j+1:}^U);$
$\quad\quad Y^S \leftarrow Y^S + y^S;$
$\quad\quad j \leftarrow |Y^U|;$
$\quad$ **end**
$\quad i \leftarrow i + 1;$
**end**
**if** $j < |Y^U|$ **then**
$\quad y^S \leftarrow \mathcal{V}(Y_{j+1:}^U);$
$\quad Y^S \leftarrow Y^S + y^S;$
**end**

---

waiting for the complete text response to be generated, ensuring low response latency that is not affected by the length of the text response. Algorithm 1 describes the above process. Additionally, since the speech decoder uses non-autoregressive modeling, the alignment corresponding to each text token $y_i^T$, specifically $A_{\lambda \cdot (i-1)+1:\lambda \cdot i}$, is generated in parallel within the chunk. Therefore, the decoding speed for generating both text and speech responses simultaneously is not significantly different from the speed of generating text response alone.

## 3 CONSTRUCTION OF SPEECH INSTRUCTION DATA: INSTRUCTS2S-200K

To train LLaMA-Omni, we need triplet data consisting of <speech instruction, text response, speech response>. However, most publicly available instruction data is in text form. Therefore, we construct speech instruction data based on existing text instruction data through the following process:

**Step 1: Instruction Rewriting**    Since speech input has different characteristics compared to text input, we rewrite the text instructions according to the following rules: (1) Add appropriate filler words (such as "hey", "so", "uh", "um", etc.) to the instructions to simulate natural speech patterns. (2) Convert non-text symbols in the instructions (such as numbers) into their corresponding spoken forms to ensure correct synthesis by TTS. (3) Modify the instructions to be relatively brief without excessive verbiage. We use the Llama-3-70B-Instruct[4] model to rewrite the instructions according to these rules. The prompt can be found in Appendix A.

**Step 2: Response Generation**    In speech interactions, existing responses from text instructions are not suitable for direct use as speech instruction responses. This is because, in text-based interactions, models tend to generate lengthy responses, using complex sentences and possibly including non-verbal elements like ordered lists or parentheses. However, in speech interactions, concise yet informative responses are typically preferred (Cho et al., 2024). Therefore, we use the Llama-3-70B-Instruct model to generate responses for speech instructions according to the following rules: (1) The response should not contain content that cannot be synthesized by the TTS model, such as parentheses, ordered lists, etc. (2) The response should be very concise and to the point, avoiding lengthy explanations. The prompt can be found in Appendix A.

**Step 3: Speech Synthesis**    After obtaining the instructions and responses suitable for speech interactions, we need to further convert them into speech using TTS models. For the instructions, to make the synthesized speech sound more natural, we use the CosyVoice-300M-SFT (Du et al., 2024) model[5], randomly selecting either a male or female voice for each instruction. For the responses, we use the VITS (Kim et al., 2021) model[6] trained on the LJSpeech (Ito & Johnson, 2017) dataset to synthesize the responses into a standard voice.

For the basic text instructions, we collect around 50K instructions from the Alpaca dataset[7] (Taori et al., 2023), which covers a wide range of topics. Additionally, we gather around 150K instructions from the UltraChat dataset[8] (Ding et al., 2023), which primarily consist of questions about the world. Note that UltraChat is a large-scale multi-turn conversation dataset, but we only select the first 150K entries and use only the first-round instruction. Using the above datasets and data processing pipeline, we ultimately obtain 200K speech instruction data, referred to as **InstructS2S-200K**. The detailed statistical information is listed in Table 1.

Table 1: Statistical information of the InstructS2S-200K dataset.

| Statistic | Value |
|---|---|
| Speech Instruction Duration | 418h |
| Speech Response Duration | 1058h |
| Avg. Speech Instruction Duration | 7.5s |
| Avg. Speech Response Duration | 19.0s |
| Avg. Text Instruction Length | 21.7 |
| Avg. Text Response Length | 39.5 |
| Avg. Unit Sequence Length | 553.6 |

## 4 EXPERIMENTS

### 4.1 EXPERIMENTAL SETUPS

**Datasets**    For the training data, we use the **InstructS2S-200K** dataset mentioned in Section 3, which includes 200K speech instruction data. To extract discrete units corresponding to the target speech, we use a pre-trained K-means quantizer[9], which has learned 1000 clusters from the HuBERT

---

[4] https://huggingface.co/meta-llama/Meta-Llama-3-70B-Instruct
[5] https://github.com/FunAudioLLM/CosyVoice
[6] https://github.com/jaywalnut310/vits
[7] https://huggingface.co/datasets/tatsu-lab/alpaca
[8] https://github.com/thunlp/UltraChat
[9] https://dl.fbaipublicfiles.com/hubert/mhubert_base_vp_en_es_fr_it3_L11_km1000.bin

features. The pretrained HiFi-GAN vocoder[10] (Kong et al., 2020; Polyak et al., 2021) is used to synthesize discrete units into waveform. For the evaluation data, we select two subsets from Alpaca-Eval[11] (Li et al., 2023): *helpful_base* and *vicuna*, as their questions are more suitable for speech interaction scenarios. We remove questions related to math and code, resulting in a total of 199 instructions. To obtain the speech version, we use the CosyVoice-300M-SFT model to synthesize the instructions into speech. We refer to this test set as **InstructS2S-Eval** in the following sections.

**Model Configuration**  We use the encoder of Whisper-large-v3 as the speech encoder, and use Llama-3.1-8B-Instruct as the LLM. The speech adapter performs a $5\times$ downsampling on the speech representations. The speech decoder consists of 2 Transformer layers with the same architecture as LLaMA, with a hidden dimension of 4096, 32 attention heads, and a feed-forward network dimension of 11008, which contains 425M parameters. The upsample factor $\lambda$ is set to 25. For the minimum unit chunk size $\Omega$ input to the vocoder, we set $\Omega = +\infty$ in the offline scenario, meaning we wait for the entire unit sequence to be generated before inputting it to the vocoder for speech synthesis. In the streaming scenario, we adjust the value of $\Omega$ within the range of [10, 20, 40, 60, 80, 100] to control the response latency of the model.

**Training**  LLaMA-Omni follows a two-stage training process. In the first stage, we train the speech adapter and the LLM with a batch size of 32 for 3 epochs. We use a cosine learning rate scheduler with the first 3% of steps for warmup, and the peak learning rate is set to 2e-5. In the second stage, we train the speech decoder, using the same batch size, number of steps, and learning rate scheduler as the first stage, but with the peak learning rate set to 2e-4. The entire training process takes approximately 65 hours on 4 NVIDIA L40 GPUs.

## 4.2 EVALUATION

Since LLaMA-Omni can generate both text and speech responses based on speech instructions, we evaluate the model's performance on two tasks: speech-to-text instruction-following (S2TIF) and speech-to-speech instruction-following (S2SIF). We use greedy search to ensure reproducible experimental results. For the S2SIF task, we further evaluate the model in two scenarios: **offline** and **streaming**. In the offline scenario, the model generates the text response first, and then synthesizes the complete speech. In the streaming scenario, the speech response is generated simultaneously with the text response. We use the following metrics to evaluate the model:

**ChatGPT Score**  To evaluate the model's ability to follow speech instructions, we use GPT-4o (OpenAI, 2024) to score the model's responses. For the S2TIF task, scoring is based on the transcribed text of the speech instructions and the model's text response. For the S2SIF task, we first transcribe the model's speech responses into text using the Whisper-large-v3 model, and then score it in the same manner as the S2TIF task. GPT-4o gives a score between 1 and 5 based on factors such as helpfulness, relevance, fluency, and suitability for speech interaction. The detailed prompt for evaluation can be found in Appendix A.

**ASR-WER**  To evaluate the alignment between text and speech responses, we use the Whisper-large-v3 model to transcribe the speech responses into text, and then calculate the Word Error Rate (WER) between the transcribed text and the text response, which is referred to as **ASR-WER**.

**UTMOS**  To evaluate the quality of the generated speech, we utilize a Mean Opinion Score (MOS) prediction model called UTMOS[12] (Saeki et al., 2022), which is capable of predicting the MOS score of the speech to assess its naturalness.

**Latency**  The response latency is a key metric for speech interaction models, referring to the time interval between the input of a speech instruction and the start of the speech response, which has a significant impact on user experience. We measure the latency on 1 NVIDIA L40 GPU.

---

[10] https://dl.fbaipublicfiles.com/fairseq/speech_to_speech/vocoder/code_hifigan/mhubert_vp_en_es_fr_it3_400k_layer11_km1000_lj/g_00500000
[11] https://github.com/tatsu-lab/alpaca_eval
[12] https://github.com/tarepan/SpeechMOS

Table 2: Results on the InstructS2S-Eval benchmark in the **offline** scenario. $\Delta$ refers to the difference in ChatGPT Score between the S2SIF and S2TIF tasks.

| Model | ChatGPT Score | | | ASR-WER $\downarrow$ | UTMOS $\uparrow$ |
| | S2TIF | S2SIF | $\Delta$ | | |
| --- | --- | --- | --- | --- | --- |
| **SpeechGPT** | 2.98 | 2.19 | 0.79 | 45.00 | 3.8958 |
| **SALMONN + Orca** | 3.44 | 3.40 | **0.04** | **3.78** | 3.8286 |
| **Qwen2-Audio + Orca** | 3.47 | 3.38 | 0.09 | 6.77 | 3.6119 |
| **LLaMA-Omni** | **3.99** | **3.47** | 0.52 | 10.82 | **3.9296** |

**Speech Rate**   To measure the speech rate of the generated speech, we use the metric Words Per Second (WPS), which represents the average number of words per second in the generated speech.

## 4.3 BASELINE SYSTEMS

We include the following speech-language models as baseline systems:

**SpeechGPT**   SpeechGPT (Zhang et al., 2023) is a speech-language model that supports both speech input and output. We use the chain-of-modality prompting adopted in the original paper for decoding, which sequentially outputs the text instruction, text response, and speech response based on the speech instruction.

**SALMONN + Orca**   SALMONN (Tang et al., 2024) is a LLM capable of accepting speech and audio inputs and responding with text, enabling it to perform the S2TIF task. For the S2SIF task, we add a TTS model called Orca[13] after SALMONN. Orca is an industrial TTS model that supports both streaming and offline speech synthesis, delivering excellent performance. This integrated system enables speech synthesis to begin concurrently with the generation of the text response.

**Qwen2-Audio + Orca**   Qwen2-Audio (Chu et al., 2024) is a powerful general-purpose audio understanding model capable of performing various audio-related tasks, including the S2TIF task. We also build a cascaded system with Qwen2-Audio and Orca to complete the S2SIF task.

When using Orca for streaming speech synthesis, we need to set a word chunk size $\Theta$, which means that speech synthesis is triggered every time $\Theta$ new words arrive. In our experiments, we varies $\Theta$ within the range of [1, 3, 5, 7, 9] to control the response latency of cascaded systems.

## 4.4 RESULTS IN THE OFFLINE SCENARIO

Table 2 presents the results on the InstructS2S-Eval benchmark in the offline scenario. For the S2TIF task, LLaMA-Omni achieves the highest ChatGPT Score, significantly outperforming the baseline systems. We attribute this to two key factors. First, our model is built upon the latest Llama-3.1-8B-Instruct model, leveraging its strong text instruction-following capabilities. Second, our InstructS2S-200K dataset effectively aligns the model with speech interaction scenarios, ensuring that its responses are both high-quality and well-suited to speech contexts. In comparison, SALMONN and Qwen2-Audio, as speech-to-text models, have not been aligned with speech interaction scenarios. As a result, their responses often include formatted content and a lot of redundant explanations, making them less suitable for such contexts.

For the S2SIF task, LLaMA-Omni still achieves the highest ChatGPT score among all models. We observe that scores for the S2SIF task are generally lower than those for the S2TIF task. This decline is primarily due to errors introduced during the speech synthesis process and the reliance on an ASR model for evaluation. The two cascaded baseline systems, which use industrial TTS models for speech synthesis, exhibit the smallest decline in scores. For the end-to-end models, LLaMA-Omni demonstrates a smaller drop in score compared to SpeechGPT, indicating that LLaMA-Omni has stronger speech generation capabilities. This can be further verified by the ASR-WER metric:

---

[13]https://github.com/Picovoice/orca

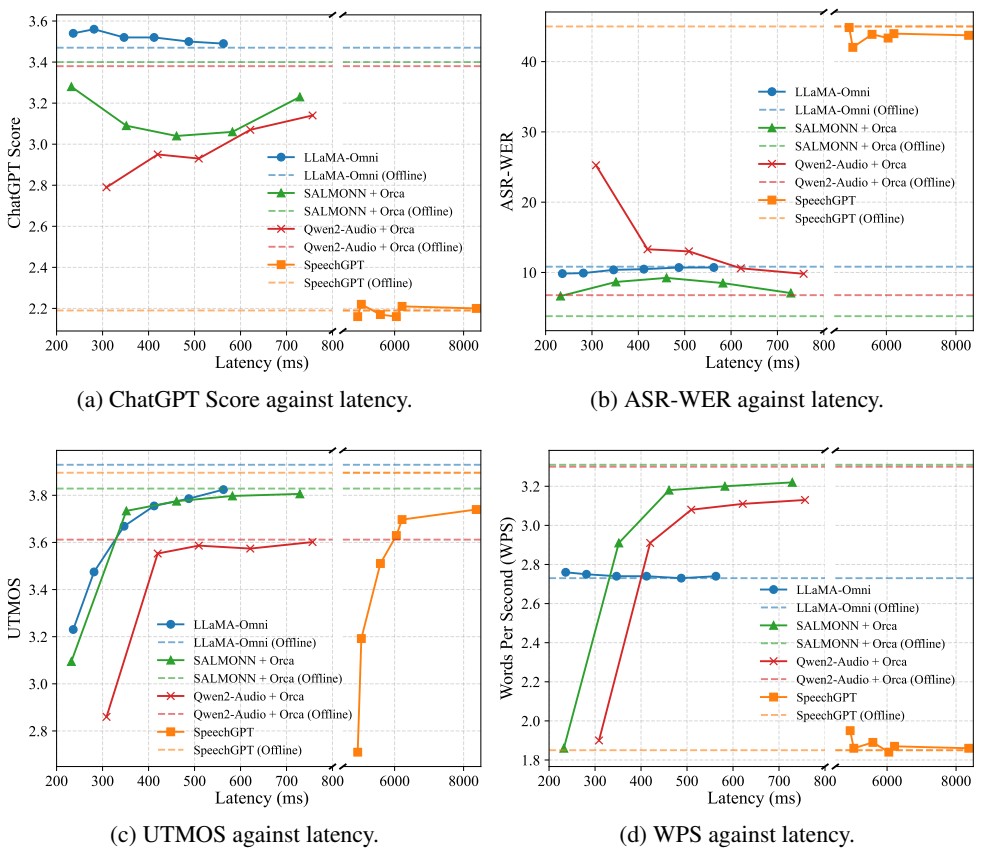

(a) ChatGPT Score against latency.

(b) ASR-WER against latency.

(c) UTMOS against latency.

(d) WPS against latency.

Figure 3: Results on the InstructS2S-Eval benchmark in the **streaming** scenario. We report the ChatGPT Score, ASR-WER, UTMOS, and WPS under different latency conditions.

the cascaded systems achieve the lowest ASR-WER, while LLaMA-Omni's ASR-WER is slightly higher but still acceptable. In contrast, SpeechGPT's ASR-WER is significantly higher, suggesting poor alignment between its speech and text responses. Although the alignment between LLaMA-Omni's speech and text responses is slightly lower than that of the cascaded systems, this is primarily due to the fact that LLaMA-Omni's speech decoder is trained on only approximately 1K hours of data, which is far less than the industrial TTS model. We believe that with more training data, LLaMA-Omni's performance could be further improved. Finally, the UTMOS metric shows that LLaMA-Omni generates satisfactory speech quality, slightly surpassing other baseline models.

### 4.5 RESULTS IN THE STREAMING SCENARIO

In the streaming scenario, the model generates speech responses simultaneously while generating text responses. For SpeechGPT and LLaMA-Omni, the response latency is controlled by adjusting the unit chunk size $\Omega$, whereas for cascaded systems, it is managed by adjusting the word chunk size $\Theta$. In Figure 3, we present the ChatGPT Score, ASR-WER, UTMOS and WPS results of all models under different latency conditions. Firstly, we examine the results of LLaMA-Omni under different latency conditions and observe that LLaMA-Omni can achieve a minimum latency of 236ms (with $\Omega = 10$), which is even lower than GPT-4o's average audio latency of 320ms. As latency increases ($\Omega$ grows larger), we notice a slight decrease in the ChatGPT Score, while ASR-WER shows a minor improvement. We believe this is mainly because the vocoder may handle shorter unit sequences more reliably than longer ones, as it is typically trained on shorter sequences. However, when $\Omega$ is smaller, the speech is divided into more segments for synthesis, which increases discontinuities in the speech. This leads to a decline in speech quality, and consequently, a decrease in the UTMOS score is observed. In addition, the speech generated by LLaMA-Omni maintains almost consistent speech rate across different latency conditions. In summary, LLaMA-Omni achieves relatively stable

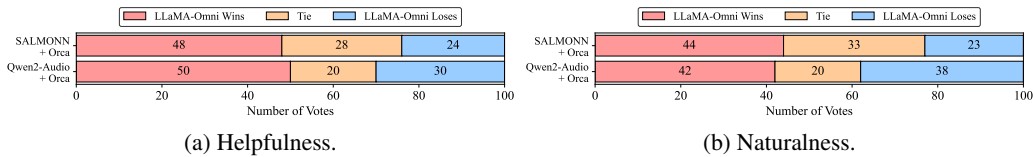

Figure 4: Results of human evaluation from the perspectives of helpfulness and naturalness.

performance under varying latency conditions. At a latency of 563ms ($\Omega = 100$), the results of all metrics are relatively close to those in the offline scenario.

Next, we compare LLaMA-Omni with other baseline systems. For SpeechGPT, its decoding latency is significantly higher than other systems ($>$4500ms) due to the sequential generation of text instructions and text responses before generating units. Additionally, the quality of its generated responses is relatively poor. For cascade systems based on streaming TTS, we observe that both Qwen2-Audio + Orca and SALMONN + Orca can achieve latencies below 300ms. However, they exhibit two notable issues. Firstly, in the streaming scenario, their ChatGPT Score and ASR-WER show significant gaps compared to the offline scenario, indicating that streaming speech synthesis at the word level tends to introducing additional errors. Secondly, as shown in Figure 3(d), when the latency is low, the overall speech rate of cascaded systems drops sharply. This is mainly because the pauses between words become more frequent, resulting in reduced naturalness and coherence of the speech. In contrast, LLaMA-Omni's streaming unit generation is performed within the end-to-end model, requiring only a cascaded vocoder to complete the streaming unit-to-waveform conversion. As a result, the overall prosody and rhythm of the speech remain almost unchanged across different latency levels. This demonstrates the advantage of end-to-end models over cascade systems based on streaming TTS. This can be more intuitively experienced by listening to the provided audio samples. We provide the numerical results of all models in the streaming scenario in Appendix C.

### 4.6 HUMAN EVALUATION

To better understand human preferences for model responses in real-time speech interaction scenarios, we further conduct a human evaluation. Specifically, we perform side-by-side comparisons between LLaMA-Omni ($\Omega = 40$, latency=347ms) and two cascaded systems: SALMONN+Orca ($\Theta = 3$, latency=352ms) and Qwen2-Audio+Orca ($\Theta = 3$, latency=420ms). For each comparison, we randomly select 20 speech instructions and collect the speech responses generated by the two models. We then invite 5 participants to evaluate all samples from two key perspectives: *helpfulness* and *naturalness*. Helpfulness evaluates whether the model follows the instructions and provides appropriate responses, while naturalness evaluates the fluency and natural quality of the generated speech. Participants compare each pair of samples rate them as win, tie, or lose for both aspects. Finally, each pairwise model comparison results in a total of 100 votes. The results, shown in Figure 5, demonstrate that LLaMA-Omni achieves a higher win rate compared to the cascade systems in both helpfulness and naturalness, confirming that LLaMA-Omni generates responses that better align with human preferences.

### 5 RELATED WORK

**Speech/Audio Language Models** With the success of language models in the field of natural language processing (Brown et al., 2020), researchers have begun exploring how to model speech or audio using language models. Early work attempted to train language models on semantic tokens or acoustic tokens of audio, enabling the generation of audio without the need for text (Lakhotia et al., 2021; Nguyen et al., 2023; Borsos et al., 2023). Furthermore, by jointly training speech tokens and text, decoder-only models like VALL-E (Wang et al., 2023b) and VioLA (Wang et al., 2023c) can perform tasks such as speech recognition, speech translation, and speech synthesis. However, the above models are not built upon LLMs. To harness the power of LLMs, many studies explore how to build speech-language models based on LLMs like LLaMA, which can be further divided into two types. The first type, represented by SpeechGPT (Zhang et al., 2023; 2024a), AudioPaLM (Rubenstein et al., 2023), and AnyGPT (Zhan et al., 2024), involves creating native multimodal speech-text models by adding speech tokens to the LLM's vocabulary and continuing pretraining using speech

and text data. However, this approach typically requires a large amount of data and substantial computational resources. The second type typically involves adding a speech encoder before the LLM and finetuning the entire model to equip it with speech understanding capabilities (Shu et al., 2023; Deshmukh et al., 2023), such as speech recognition (Fathullah et al., 2024a; Yu et al., 2024; Ma et al., 2024b; Hono et al., 2024), speech translation (Wu et al., 2023; Wang et al., 2023a; Chen et al., 2024), or other general speech-to-text tasks (Chu et al., 2023; Tang et al., 2024; Chu et al., 2024; Fathullah et al., 2024b; Das et al., 2024; Hu et al., 2024). However, these approaches typically focus only on speech or audio understanding without the ability to generate them. Compared to previous work, LLaMA-Omni equips the LLM with both speech understanding and generation capabilities, enabling it to perform general speech instruction-following tasks. Recently, some contemporary works such as Mini-Omni (Xie & Wu, 2024) and Moshi (Défossez et al., 2024) have also focused on speech interactions with LLMs, aiming to improve the quality of speech response by simultaneously generating both text and speech. Compared to them, our advantages include: (1) LLaMA-Omni is built upon the latest LLM Llama-3.1-8B-Instruct model, which has stronger reasoning capabilities; (2) We utilize CTC to adaptively learn the alignment between speech and text responses, eliminating the need to pre-align speech and text during training; (3) Our training only uses 200K data samples, which is several orders of magnitude less than theirs, and the training requires only 4 GPUs for 3 days, making the training cost significantly lower.

**Simultaneous Generation**   Streaming generation aims to begin producing output before the entire input is received. This capability is crucial for maintaining synchronization between speakers and listeners in various scenarios, such as streaming speech recognition and simultaneous interpretation. In the case of large language models, having a streaming speech synthesis component can significantly reduce latency between the model and its users. Popular streaming generation methods fall into three main categories: monotonic-attention-based methods (Raffel et al., 2017), CTC-based methods (Graves et al., 2006b), and Transducer-based methods (Graves, 2012). Monotonic-attention-based methods modify the traditional attention-based sequence-to-sequence framework (Bahdanau, 2014) to support streaming generation. These methods rely on an external module to manage the READ/WRITE policy, which can be either fixed (e.g., Wait-k (Ma et al., 2018)) or adaptive (e.g., MMA (Ma et al., 2019), EDAtt (Papi et al., 2022), Seg2Seg (Zhang & Feng, 2024)). CTC-based methods add a blank symbol to the target vocabulary to represent a WAIT action. Streaming inference is achieved by removing adjacent repetitive tokens and blank symbols, which has proven effective in simultaneous interpretation and streaming speech synthesis (Ma et al., 2023; Zhang et al., 2024b; Ma et al., 2024a). Transducer-based methods are designed to bridge the gap between the non-autoregressive nature of CTC-based methods and the autoregressive dependency between target tokens. These approaches introduce an additional predictor to capture token dependencies, and their variants have also shown strong performance in simultaneous interpretation (Liu et al., 2021; Tang et al., 2023) and streaming speech synthesis (Chen et al., 2021). For the streaming TTS task, many studies employ straightforward lookahead strategies built upon models like Tacotron 2 (Shen et al., 2018), such as waiting for a few future words (Ma et al., 2020; Stephenson et al., 2020) or leveraging a language model to predict several future words (Saeki et al., 2021a;b; Liu et al., 2022), enabling improved streaming speech synthesis. Dekel et al. (2024) achieves streaming speech synthesis simultaneously with the output stream of LLMs by employing two cascaded streamable models that sequentially generate phonemes and speech in real-time.

## 6  CONCLUSION

In this paper, we propose an innovative model architecture, LLaMA-Omni, which enables low-latency and high-quality speech interaction with LLMs. LLaMA-Omni is built upon the latest Llama-3.1-8B-Instruct model, with the addition of a speech encoder for speech understanding and a streaming speech decoder that can generate both text and speech responses simultaneously. To align the model with speech interaction scenarios, we construct a speech instruction dataset InstructionS2S-200K, which contains 200K speech instructions along with the speech responses. Experimental results show that, compared to previous speech-language models, LLaMA-Omni delivers superior responses in both content and style, with a response latency as low as 236ms. Moreover, training LLaMA-Omni requires less than 3 days on 4 GPUs, enabling rapid development of speech interaction models based on the latest LLMs. In the future, we plan to explore enhancing the expressiveness of generated speech responses and improving real-time interaction capabilities.

ACKNOWLEDGMENTS

We thank all the anonymous reviewers for their insightful and valuable comments. This paper is supported by National Natural Science Foundation of China (Grant No.62376260).

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

# A  PROMPT

**Prompt template of LLaMA-Omni**

`<|begin_of_text|><|start_header_id|>system<|end_header_id|>`
You are a helpful language and speech assistant. You are able to understand the speech content that the user provides, and assist the user with a variety of tasks using natural language. `<|eot_id|>`
`<|start_header_id|>user<|end_header_id|>`
**\<speech\>**
Please answer the questions in the user's input speech. `<|eot_id|>`
`<|start_header_id|>assistant<|end_header_id|>`

**Prompt for ChatGPT Scoring (Model: GPT-4o)**

I need your help to evaluate the performance of several models in a speech interaction scenario. The models receive the user's speech input and respond with speech output. For evaluation purposes, both the user's speech input and the model's speech response have been transcribed into text using Automatic Speech Recognition (ASR). Your task is to rate the model's responses based on the provided user input transcription [Instruction] and the model's output transcription [Response]. Please consider factors such as helpfulness, relevance, fluency, and suitability for speech interaction in your evaluation, and provide a single score on a scale from 1 to 5.

Below are the transcription of user's instruction and models' response:
### [Instruction]: **{instruction}**
### [Response]: **{response}**

After evaluating, please output the scores in JSON format: {score: ...}. You don't need to provide any explanations.

**Prompt for Instruction Rewriting (Model: Llama-3-70B-Instruct)**

Below is an instruction data containing the user's instruction. I would like to generate a speech version of this instruction for training a large language model that supports speech input. Therefore, please rewrite my instruction data according to the following requirements:

1. Modify the instruction to simulate human speech, adding fillers as appropriate (but not too many 'you know', 'like', etc.).
2. The question should not contain content that cannot be synthesized by the TTS model. Numbers should be written in English words rather than Arabic numerals.
3. The question should be relatively brief without excessive verbiage.

[instruction]: **{instruction}**

Please output in JSON format as follows: {"question": {question}}.

**Prompt for Response Generation (Model: Llama-3-70B-Instruct)**

Below is the transcribed text of a user's speech query. Please provide a response to this question, which will be converted to speech using TTS. Please follow these requirements for your response:

1. Your response should not contain content that cannot be synthesized by the TTS model, such as parentheses, ordered lists, etc. Numbers should be written in English words rather than Arabic numerals.
2. Your response should be very concise and to the point, avoiding lengthy explanations.

[instruction]: **{instruction}**

Please output in JSON format as follows: {"response": {response}}.

# B  CASE STUDY

To intuitively understand the differences in responses from different models, we provide an example in Table 3. It can be observed that the response of Qwen2-Audio are quite lengthy and include elements such as line breaks and parentheses that cannot be synthesized into speech. The response

Table 3: Responses from different models for the instruction: "How do I wrap a present neatly?"

| Model | Response |
|-------|----------|
| Qwen2-Audio | Wrapping a present neatly can be a bit tricky, but with some practice and patience, you can master the art of wrapping presents beautifully. Here are some steps to follow:\n\n1. Start with a clean, flat surface: Find a clean, flat surface where you can work on wrapping your present. A large table or countertop works well.\n\n2. Gather your supplies: You will need wrapping paper, gift wrap scissors, a ruler or measuring tape, and any other decorations you want to use (such as ribbon, bows, or stickers).\n\n3. Measure the gift: Use the ruler or measuring tape to measure the width and length of the gift. Make sure to add a little extra space around the edges for wrapping.\n\n4. Cut the wrapping paper: Cut the wrapping paper to the size you measured. If you have multiple pieces of wrapping paper, layer them together so that they fit neatly around the gift.\n\n5. Fold the paper: Begin by folding one of the sides of the wrapping paper over the gift, crease it well, and then fold the other side over the top. This creates a simple, yet attractive fold.\n\n6. Add decorations: If you like, you can add decorations such as ribbon, bows, or stickers to the gift. Simply attach them to the top of the gift using glue or double-sided tape.\n\n7. Wrap the gift: Starting at the bottom of the gift, place a piece of wrapping paper over it. Bring the left side of the paper up over the gift, tuck it in tightly, and press it down. Repeat this process on the right side of the gift, bringing the right side of the paper up over the gift, tucking it in tightly, and pressing it down.\n\n8. Finish the edges: Once you have wrapped the gift, trim off any excess paper along the edges using the gift wrap scissors. You can also add additional touches such as a bow or decorative label to complete the look.\n\nRemember, the key to wrapping a present neatly is to take your time and pay attention to detail. With practice, you'll be able to create beautiful wrapping paper presentations every time! |
| SALMONN | To wrap a present neatly, you can start by laying out a clean and flat surface to work on. Then, place the present on the surface and wrap it in a piece of wrapping paper that is slightly larger than the present. Fold the paper in half and then in half again, so that the top and bottom flaps meet in the middle. Then, fold the left and right flaps in towards the center, so that they meet at the top and bottom of the present. Finally, seal the present by pressing the flaps together and using some tape to secure them. |
| SpeechGPT | Wrapping a present neatly requires a few simple steps. First, you should fold the paper in half and then unfold it. Then, you should place the present in the center of the paper and fold the paper around it. Finally, you should tie the ends of the paper with a ribbon or string. |
| LLaMA-Omni | To wrap a present neatly, start with a flat surface, place the gift in the center, and fold one edge of the wrapping paper over the gift, crease it, and tape it down. Repeat this process with the remaining edges, making sure to align them evenly and smooth out any air bubbles. |

from SALMONN is also a bit long. The style of SpeechGPT's response is more appropriate for speech interaction scenarios, but the amount of information contained in its responses is less. In contrast, the response given by LLaMA-Omni is more detailed and helpful while maintaining a concise style, outperforming previous models in speech interaction scenarios.

## C NUMERICAL RESULTS IN THE STREAMING SCENARIO

We present the numerical results of all models in streaming scenarios in Tables 4, 5, 6, and 7.

Table 4: Numerical results of LLaMA-Omni in the streaming scenario.

| $\Omega$ | Latency (ms) | | | ChatGPT Score | ASR-WER | UTMOS | WPS |
|---------|------|----------|-------|---------------|---------|-------|-----|
|         | LLM  | Vocoder  | Total |               |         |       |     |
| 10 | 206.03 | 30.15 | 236.18 | 3.54 | 9.84 | 3.2304 | 2.76 |
| 20 | 236.18 | 45.23 | 281.41 | 3.56 | 9.91 | 3.4748 | 2.75 |
| 40 | 301.51 | 45.23 | 346.73 | 3.52 | 10.37 | 3.6688 | 2.74 |
| 60 | 361.81 | 50.25 | 412.06 | 3.52 | 10.47 | 3.7549 | 2.74 |
| 80 | 432.16 | 55.28 | 487.44 | 3.50 | 10.70 | 3.7858 | 2.73 |
| 100 | 497.49 | 65.33 | 562.81 | 3.49 | 10.71 | 3.8242 | 2.74 |
| Offline | 1542.71 | 211.06 | 1753.77 | 3.47 | 10.82 | 3.9296 | 2.73 |

Table 5: Numerical results of SpeechGPT in the streaming scenario.

| Ω | Latency (ms) | | | ChatGPT Score | ASR-WER | UTMOS | WPS |
|---|---|---|---|---|---|---|---|
| | LLM | Vocoder | Total | | | | |
| 10 | 4899.50 | 30.15 | 4929.65 | 2.16 | 44.85 | 2.7099 | 1.95 |
| 20 | 5005.03 | 30.15 | 5035.18 | 2.22 | 42.03 | 3.1920 | 1.86 |
| 40 | 5547.74 | 40.20 | 5587.94 | 2.17 | 43.88 | 3.5106 | 1.89 |
| 60 | 6005.03 | 45.23 | 6050.25 | 2.16 | 43.35 | 3.6293 | 1.84 |
| 80 | 6160.80 | 55.28 | 6216.08 | 2.21 | 43.98 | 3.6970 | 1.87 |
| 100 | 8301.51 | 65.33 | 8366.83 | 2.20 | 43.74 | 3.7397 | 1.86 |
| Offline | 17919.60 | 170.85 | 18090.45 | 2.19 | 45.00 | 3.8958 | 1.85 |

Table 6: Numerical results of SALMONN + Orca in the streaming scenario.

| Θ | Latency (ms) | | | ChatGPT Score | ASR-WER | UTMOS | WPS |
|---|---|---|---|---|---|---|---|
| | LLM | TTS | Total | | | | |
| 1 | 212.45 | 19.71 | 232.16 | 3.28 | 6.64 | 3.0947 | 1.86 |
| 3 | 316.59 | 35.08 | 351.67 | 3.09 | 8.65 | 3.7338 | 2.91 |
| 5 | 428.79 | 32.08 | 460.87 | 3.04 | 9.23 | 3.7750 | 3.18 |
| 7 | 536.47 | 45.90 | 582.37 | 3.06 | 8.49 | 3.7972 | 3.20 |
| 9 | 659.57 | 69.38 | 728.95 | 3.23 | 7.07 | 3.8060 | 3.22 |
| Offline | 4274.48 | 1049.59 | 5324.07 | 3.40 | 3.78 | 3.8286 | 3.31 |

Table 7: Numerical results of Qwen2-Audio + Orca in the streaming scenario.

| Θ | Latency (ms) | | | ChatGPT Score | ASR-WER | UTMOS | WPS |
|---|---|---|---|---|---|---|---|
| | LLM | TTS | Total | | | | |
| 1 | 289.46 | 19.15 | 308.61 | 2.79 | 25.25 | 2.8597 | 1.90 |
| 3 | 381.66 | 38.19 | 419.85 | 2.95 | 13.30 | 3.5529 | 2.91 |
| 5 | 470.55 | 38.64 | 509.19 | 2.93 | 13.00 | 3.5865 | 3.08 |
| 7 | 568.22 | 52.95 | 621.17 | 3.07 | 10.59 | 3.5739 | 3.11 |
| 9 | 675.55 | 81.02 | 756.57 | 3.14 | 9.81 | 3.6016 | 3.13 |
| Offline | 7062.93 | 2361.49 | 9424.42 | 3.38 | 6.77 | 3.6119 | 3.30 |

