# OpenReview forum: "LLaMA-Omni: Seamless Speech Interaction with Large Language Models"
_ICLR.cc/2025/Conference — ICLR 2025 Poster_

### Official Review · Reviewer_4A7w · 2024-10-19

**Soundness:** 4
**Presentation:** 4
**Contribution:** 3
**Rating:** 8
**Confidence:** 4

**Summary:**

This paper introduces LLama-Omni, a novel end-to-end architecture that enables real-time interaction with LLMs through speech. Llama-Omni has a pretrained speech encoder, a speech adaptor, an LLM, and a streaming speech decoder. Speech is fed continuously into the LLM, while text is generated discretely. Then, each text unit is replicated a fixed number of times and aligned with HuBERT k-means units using non-autoregressive CTC. Once the units accumulate to a preset number, they form a chunk and feed to a vocoder for speech reconstruction. This non-autoregressive architecture supports simultaneous text and speech generation from spoken instructions with very low latency, while achieving very low training overhead.

**Strengths:**

- LLama-Omni carries out a novel non-autoregressive architecture for simultaneous text and speech generation, providing a fresh approach among recent GPT-4o-like models. It achieves a sufficiently low latency of 226ms and high-quality speech interaction, making it suitable for real-time scenarios.

- LLaMA-Omni can be integrated with any open-source LLM at minimal training cost, requiring only 4 GPUs and less than 3 days to train. This makes it accessible for academic research, allowing more institutions and researchers to use high-quality, speech-interactive LLMs.

- One notable strength is that LLaMA-Omni largely preserves the intelligence of pre-trained LLMs.

**Weaknesses:**

- One limitation is that the LLM outputs text only. While this preserves its reasoning abilities, it losses richer features such as emotional nuance and speaker identity.

- As the authors mentioned in Section 4.4, the cascaded use of VITS TTS in SALMONN and Qwen2-Audio may introduce errors affecting S2SIF and alignment due to propagated TTS inaccuracies, making this comparison less of an apple-to-apple evaluation.

**Questions:**

- Is there a risk of audio artifacts, such as "popping", at the boundaries when using the vocoder to generate speech chunks? If so, how does the model mitigate these issues?

- Could you provide more detailed comparisons between LLaMA-Omni and other speech-interactive LLMs to evaluate whether introducing speech as an input/output modality affects the model’s reasoning and language comprehension abilities? Specifically, does enabling speech interaction in a text-based LLM reduce its core language understanding and intelligence? Personally, I evaluated another GPT-4o-like model, mini-Omni, and found that their method to introduce the speech modality significantly reduced the language model's core intelligence. Although the authors repeatedly claimed in the paper that their method could "retain the original model’s language capabilities with minimal degradation," they provided no evaluation to substantiate this claim.

---

> ### Author Response · Authors · 2024-11-23
> **Author Response (1/1)**
>
> Thank you for providing your valuable and constructive feedback! Please find below the responses to each comment.
>
> ---
>
> **[Q1] Losses richer features such as emotional nuance and speaker identity.**
>
> Thank you for your question. As you mentioned, our speech is not directly generated by the LLM but instead by an additional speech decoder. While this approach may somewhat weaken the speech modeling capabilities, it preserves the fundamental reasoning abilities to a greater extent, which we believe is a more essential and foundational aspect at this stage.
>
> Additionally, since our speech generation still follows an end-to-end modeling approach, the model retains the potential to perceive information beyond the content of the input speech, such as emotion. However, enabling the model to generate speech with emotions or other richer characteristics requires further exploration, which we plan to address in future work.
>
> ---
>
> **[Q2] The cascaded use of VITS TTS in SALMONN and Qwen2-Audio may introduce errors affecting S2SIF and alignment due to propagated TTS inaccuracies.**
>
> Thank you for your question. To ensure a fairer comparison, we replaced the TTS component of the cascaded systems with a industrial streaming TTS model, Orca, and conducted more extensive comparative experiments across different latency settings. The relevant experimental results have been updated in the revised PDF. If you are interested, please refer to the updated paper for details.
>
> ---
>
> **[Q3] "Popping" at the boundaries.**
>
> Yes, we have also observed that popping artifacts can occur between audio segments synthesized from different unit chunks. We believe that addressing this issue requires enhancing the vocoder's streaming synthesis capabilities, such as employing lookahead strategies during synthesis to prevent discontinuities at the end frames. Additionally, our current approach synthesizes each unit chunk independently before concatenation. Incorporating the preceding unit chunk during synthesis could potentially alleviate this issue as well.
>
> ---
>
> **[Q4] Language comprehension abilities.**
>
> Thank you for your question. Following your suggestion, we tested our base model Llama-3.1-8B-Instruct and LLaMA-Omni on the MMLU benchmark. The results are presented in the table below:
>
> | Models                | Humanities | Social Science | STEM      | Other     | Avg.      |
> | --------------------- | ---------- | -------------- | --------- | --------- | --------- |
> | Llama-3.1-8B-Instruct | **73.06**  | **78.39**      | **59.25** | **70.60** | **69.02** |
> | LLaMA-Omni            | 70.21      | 74.99          | 57.30     | 70.27     | 66.93     |
>
> We observed that the trained model exhibited a slight degradation in its language comprehension capabilities, although it retained most of its original abilities. We believe this is primarily because no text instruction data was used during the training phase, leading to reduced performance when evaluated on text instruction datasets. This issue might be addressed using simple data replay techniques—by incorporating some text instructions during the speech instruction fine-tuning process. We plan to explore this approach in future work.

---

> ### Comment · Reviewer_4A7w · 2024-11-24
> **Thank you for the response**
>
> Thank the authors for their huge efforts in revising the paper. I appreciate the responses that address most of my concerns. I believe the experiments are generally convincing and comprehensive. More importantly, another notable strength is that this paradigm largely preserves the intelligence of pre-trained LLMs.
>
> My rating leans towards accept.

---

> > ### Author Response · Authors · 2024-11-24
> > **Thank you for your response!**
> >
> > Thank you for your quick response and positive feedback. We're glad to hear that your concerns have been resolved, and we truly appreciate you raising the score. Have a nice day!

---

### Official Review · Reviewer_DXz7 · 2024-10-23

**Soundness:** 3
**Presentation:** 3
**Contribution:** 2
**Rating:** 6
**Confidence:** 4

**Summary:**

This work presents an end-to-end spoken dialogue model LLAMA-Omini, which integrates a pretrained speech encoder, a speech adaptor, an LLM, and a streaming TTS module. The model can directly understand continuous speech features and can simultaneously generate text and speech responses directly with low latency. To achieve alignment between different modules, this work also proposes a pipeline for constructing speech-to-speech dialogue data, with a style that better matches the characteristics of speech interaction scenarios.

**Strengths:**

1. The paper is well-written and presents a promising approach to seamless speech interaction with LLMs. The reported response latency of 226ms is impressive and demonstrates the value for real-world applications.

2. The authors have made a valuable contribution by creating the InstructS2S-200K dataset, which is tailored to speech interaction scenarios. The core motivation for building this dataset is that "in speech interactions, concise yet informative responses are typically preferred", which has not been considered in previous work.

**Weaknesses:**

1. Although this work presents a seamless spoken dialogue model, the choice of connecting a LLM with a CTC-based streaming TTS module is not well-motivated.
    - Autoregressive TTS models naturally support streaming decoding, but this paper does not discuss or compare with the traditional autoregressive decoding.

    - The CTC-based streaming TTS module incorporates ideas from StreamSpeech [1] and achieves streaming output by segmenting generated units into chunks. However, it is not evident what benefits the end-to-end model offers over a straightforward cascaded solution where the text response is segmented into chunks and then fed into an offline TTS model.

2. Another contribution of this work is the constructed dataset InstructS2S-200K, but there is a lack of sufficient experiments to demonstrate the advantages of this dataset.

   - First of all, the speech input in this dataset is synthesized using CosyVoice-300M-SFT, which can only synthesize standard speech from limited speakers. This may lead to the model only being able to handle tts-synthesized input speech, lacking speaker generalization. Therefore, in order to demonstrate the practical value of this dataset, evaluation on human recorded speech is necessary. For example, ASR and ST performance in academic dataset (LibriSpeech, CoVoST), or evaluation on general benchmarks (such as AIRBench [2]) is necessary to demonstrate the model's understanding of real speech. But this work is only evaluated on their TTS-synthesized InstructS2S-Eval.

   - One advantage of this dataset is that the response is accurate and concise, which is more suitable for the speech interaction scenario. However, in order to prove this statement, human evaluation is very necessary to demonstrate that humans would prefer shorter replies. In Table 1, the author uses GPT4 as a judge, but in Appendix A, we can see that the author has incorporated the need for short and accurate responses into the GPT4 evaluation prompt. This can only prove that the model's responses are accurate and concise, but cannot prove that they are more in line with human preferences.

3. There are some unconvincing evaluation metrics and results analysis.

   - In Table 1, a biased prior has been introduced in the evaluation prompt of GPTscore (see 2.b), therefore the content and style scores are not reasonable.

   - In Table 1, the pipeline models SALMOON/Qwen2-Audio + TTS have very poor alignment scores. Although the authors explained in Line 392-393 that the responses of these two models contain characters that cannot be synthesized into speech, we can see in the case study in Table 5 that SALMMON does not have any unsynthesizable characters, and Qwen2-Audio only has \n that is unsynthesizable. I speculate that the very poor alignment score here is due to the fact that the TTS model used only supports synthesizing speech for around 10 seconds, while SALMOON and Qwen2-Audio generate very long responses that exceed the maximum supported length of the TTS model. I believe the authors need to provide a detailed description of how they used the TTS model, or to segment long text responses into shorter chunks to fed into TTS model to ensure fairness in comparison.

   - Table 4 reports the total time for completing a single response, and it is quite apparent that shorter responses result in shorter completion times. However, this does not provide any new insights or perspectives to demonstrate the advantages of this work.

4. The author emphasizes that LLAMA-Omini is an interactive dialogue model, or a GPT-4o-like model, and therefore needs to demonstrate its ability for multi-turn conversations. However, evaluations have only been conducted on single-turn responses. I think that quantitative metrics or qualitative demos are necessary.

[1] Zhang, Shaolei, et al. StreamSpeech: Simultaneous Speech-to-Speech Translation with Multi-task Learning. ACL 2024.

[2] Yang, Qian, et al. AIR-Bench: Benchmarking Large Audio-Language Models via Generative Comprehension. arxiv 2402.07729.

**Questions:**

1. Could the author please emphasize the contribution of this article?
   - If it is a streaming TTS module, there is a lack of comparison with other streaming generation methods (autoregressive TTS models)
   - if it is the speech instruction dataset, there is a lack of human evaluation to prove its greater alignment with human preference, and the limitations of TTS synthesized speech input also restrict its application
   - if it is an open-source seamless spoken dialogue model, evaluation on real speech benchmarks is also needed.

---

> ### Author Response · Authors · 2024-11-23
> **Author Response (1/n)**
>
> Thank you for providing your valuable and constructive feedback! Please find below the responses to each comment.
>
> ---
>
> **[Q1] Comparison with autoregressive TTS models**
>
> We respectfully disagree with your statement that "Autoregressive TTS models naturally support streaming decoding." Autoregressive TTS models with decoder-only architectures (e.g., CosyVoice, VALL-E) can perform streaming speech generation **when the input text is complete**. However, **when the input text arrives incrementally, such as in the case of LLM output streams, we argue that autoregressive TTS models cannot inherently support streaming decoding**.
>
> For example, when the first word $ X_1 $ arrives in the text stream, an autoregressive TTS model decodes it into $[Y_1, Y_2, Y_3]$. When the second word $ X_2 $ arrives, there are two possible strategies:
>
> 1. Append $ X_2 $ after $ X_1 $, which changes the context and necessitates re-encoding $[Y_1, Y_2, Y_3]$. This results in the need to re-encode all target speech positions each time a new word arrives.
> 2. Append $ X_2 $ after $ Y_3 $, feeding $[X_1, Y_1, Y_2, Y_3, X_2]$ into the model. This introduces a mismatch with the training paradigm, as the model is typically trained with input-output sequences in the form $[X_1, ..., X_n, Y_1, ..., Y_m]$.
>
> Thus, we believe that autoregressive TTS models are inherently unsuited for streaming decoding in scenarios where the input text is incrementally provided. **In contrast, our non-autoregressive model decouples inputs and outputs, ensuring that outputs do not need to be re-fed into the model. This design makes it inherently compatible with streaming scenarios.**
>
> ---
>
> **[Q2] It is not evident what benefits the end-to-end model offers over a straightforward cascaded solution where the text response is segmented into chunks and then fed into an offline TTS model.**
>
> Thank you for your question. We have updated two cascaded baseline systems with a streaming TTS model, which aligns with your suggestion of splitting LLM's text outputs into chunks and feeding them into a streaming TTS model. The following experimental results will demonstrate the superiority of our method compared to the cascaded system.
>
> ---
>
> **[Q3] Lacking speaker generalization.**
>
> Thank you for your suggestion. As you pointed out, the speech instructions in our training set were synthesized using CosyVoice, which includes only two standard voices (male and female). **However, since our speech encoder (whisper) remains frozen throughout training, its inherent ability to handle multi-speaker speech remains intact.**
>
> To verify this, we synthesized a new training and evaluation dataset using the [**fish-speech-1.4**](https://huggingface.co/fishaudio/fish-speech-1.4) model, with a randomly selected voice for each synthesis. **This ensured that all data samples have unique and non-overlapping voices.** We then trained a new model on this dataset while keeping all other settings unchanged. Finally, we evaluated the model on two test sets: the **single-speaker test set synthesized by CosyVoice** (the same test set used in other experiments) and the **multi-speaker test set synthesized by fish-speech-1.4**. The results are presented in the table below.
>
> | ID   | Training   | Evaluation | S2TIF ChatGPT Score |
> | ---- | ---------- | ---------- | ------------------- |
> | Exp1 | CosyVoice  | CosyVoice  | 3.99                |
> | Exp2 | CosyVoice  | FishSpeech | 4.07                |
> | Exp3 | FishSpeech | CosyVoice  | 3.96                |
> | Exp4 | FishSpeech | FishSpeech | 4.19                |
>
> Comparing Exp1 and Exp2, we observe that **even the model trained on only two standard voices performs reasonably well on the multi-speaker test set**, with some improvement observed (possibly due to differences in synthesis accuracy between CosyVoice and FishSpeech). However, comparing Exp2 and Exp4, we find that **the model trained on multi-speaker data performs better on the multi-speaker test set** (4.07 vs. 4.19). This demonstrates the benefits of incorporating voice diversity during training.
>
> In summary, we agree with your point that training with multi-speaker data can enhance the model's robustness. However, **we believe that the current model is already sufficient to handle real-world scenarios effectively**.

---

> ### Author Response · Authors · 2024-11-23
> **Author Response (2/n)**
>
> **[Q4] Evaluation on human recorded speech.**
>
> We did not test on datasets such as LibriSpeech, CoVoST, or AIRBench because our model is designed for scenarios that differ from these tasks. Specifically, our model generates responses based solely on speech instructions, without requiring any text instructions. Tasks such as speech recognition, speech translation, or speech-based question answering, however, necessitate the model to simultaneously process speech input and text instructions, which is not encountered during our model’s training.
>
> To address this, we conducted a fine-tuning experiment on LLaMA-Omni using the LibriSpeech `train-clean-100h` dataset. With just one epoch of fine-tuning (892 steps), the model achieved Word Error Rates (WER) of 3.29% on the `test-clean` set and 6.88% on the `test-other` set. From the experimental results, we conclude that LLaMA-Omni inherently possesses the capability to process various types of speech. Fine-tuning merely serves to teach the model how to follow speech recognition instructions.
>
> ---
>
> **[Q5] Unconvincing evaluation metrics and results analysis.**
>
> We sincerely thank you for pointing out potential issues in our experimental results. Based on your suggestions, we conducted a comprehensive review of our experiments and identified several key issues, which we have thoroughly corrected:
>
> 1. **ASR-WER/CER**: Regarding your observation of unusually high ASR-WER/CER scores of the Qwen2-Audio+TTS Model, we found this was due to an oversight. When calculating ASR-WER/CER, the Whisper model incorrectly truncated the speech to 30 seconds, resulting in the loss of content after the 30-second mark, which led to incorrect ASR-WER/CER scores. **We have corrected this issue and conducted a complete re-evaluation.**
> 2. **ChatGPT Score**: The evaluation prompt we previously used may have introduced bias. **We have removed all potentially bias-inducing priors from the prompt** and simply instructed the model to give a single score based on helpfulness, relevance, fluency, and suitability for speech interaction, without providing any explanations that might introduce bias (details in Appendix A of the revised PDF). **We also conducted human evaluations following your suggestions**, and the results are presented below.
> 3. **TTS Model**: Apologies for the previous unclear explanation, which may have caused misunderstandings. **The VITS model we used is integrated within the [Coqui TTS framework](https://github.com/coqui-ai/TTS), which performs text segmentation before speech synthesis to ensure accurate synthesis even for long texts**. In fact, the speech responses in our training dataset were also synthesized using this model. However, to achieve a more comprehensive and meaningful comparison, especially in the streaming scenario, **we have replaced the cascaded baseline systems with Qwen2-Audio/SALMONN + [Orca](https://github.com/Picovoice/orca), where Orca is an industrial TTS model that supports both streaming and offline speech synthesis**. When using Orca for streaming speech synthesis, we need to set a **word chunk size** $\Theta$, which means that speech synthesis is triggered every time $\Theta$ new words arrive. In our experiments, we varies $\Theta$ within the range of $[1, 3, 5, 7, 9]$ to control the response latency of cascaded systems.
>
> **We conducted a comprehensive re-evaluation of all models in both offline and streaming scenarios**. Specifically, we evaluate the model using the following metrics:
>
> 1. **ChatGPT Score [Updated]**: We use GPT-4o to rate the model's responses. Following the suggestions from reviewers ppgU and DXz7, we have revised the evaluation prompt (details in Appendix A of the revised PDF) to remove potential prior biases.
> 2. **ASR-WER [Unchanged]**: This metric evaluates the alignment between text and speech responses. We have corrected the errors that occurred during the previous ASR process.
> 3. **UTMOS [Unchanged]**: This metric assesses the quality of the generated speech.
> 4. **WPS (Words Per Second) [New]**: This metric measures the speaking rate of the generated speech, which is calculated as the average number of words per second in the generated speech.
> 5. **Latency [Updated]**: Latency refers to the **total delay** between the user's speech input and the moment they hear the speech response. This can be further divided into the **decoding latency of the LLM** and **the latency introduced by the Vocoder or TTS model**.

---

> ### Author Response · Authors · 2024-11-23
> **Author Response (3/n)**
>
> The numerical results under different latency conditions for the cascade systems (SALMONN + Orca, Qwen2-Audio + Orca) and LLaMA-Omni are as follows:**[SALMONN + Orca]**
>
> | $\mathbf{\Theta}$ | Latency-LLM (ms) | Latency-TTS (ms) | Latency-Total (ms) | ChatGPT Score | ASR-WER | UTMOS   | WPS  |
> |--------------------|----------|----------|------------|---------------|---------|---------|------|
> | 1                 | 212.45   | 19.71    | 232.16     | 3.28          | 6.64    | 3.0947  | 1.86 |
> | 3                 | 316.59   | 35.08    | 351.67     | 3.09          | 8.65    | 3.7338  | 2.91 |
> | 5                 | 428.79   | 32.08    | 460.87     | 3.04          | 9.23    | 3.7750  | 3.18 |
> | 7                 | 536.47   | 45.90    | 582.37     | 3.06          | 8.49    | 3.7972  | 3.20 |
> | 9                 | 659.57   | 69.38    | 728.95     | 3.23          | 7.07    | 3.8060  | 3.22 |
> | Offline           | 4274.48  | 1049.59  | 5324.07    | 3.40          | 3.78    | 3.8286  | 3.31 |
>
> **[Qwen2-Audio + Orca]**
>
> | $\mathbf{\Theta}$ | Latency-LLM (ms) | Latency-TTS (ms) | Latency-Total (ms) | ChatGPT Score | ASR-WER | UTMOS   | WPS  |
> |--------------------|----------|----------|------------|---------------|---------|---------|------|
> | 1                 | 289.46   | 19.15    | 308.61     | 2.79          | 25.25   | 2.8597  | 1.90 |
> | 3                 | 381.66   | 38.19    | 419.85     | 2.95          | 13.30   | 3.5529  | 2.91 |
> | 5                 | 470.55   | 38.64    | 509.19     | 2.93          | 13.00   | 3.5865  | 3.08 |
> | 7                 | 568.22   | 52.95    | 621.17     | 3.07          | 10.59   | 3.5739  | 3.11 |
> | 9                 | 675.55   | 81.02    | 756.57     | 3.14          | 9.81    | 3.6016  | 3.13 |
> | Offline           | 7062.93  | 2361.49  | 9424.42    | 3.38          | 6.77    | 3.6119  | 3.30 |
>
> **[LLaMA-Omni]**
>
> | $\mathbf{\Omega}$ | Latency-LLM (ms) | Latency-Vocoder (ms) | Latency-Total (ms) | ChatGPT Score | ASR-WER | UTMOS   | WPS  |
> |--------------------|----------|--------------|------------|---------------|---------|---------|------|
> | 10                | 206.03   | 30.15        | 236.18     | 3.54          | 9.84    | 3.2304  | 2.76 |
> | 20                | 236.18   | 45.23        | 281.41     | 3.56          | 9.91    | 3.4748  | 2.75 |
> | 40                | 301.51   | 45.23        | 346.73     | 3.52          | 10.37   | 3.6688  | 2.74 |
> | 60                | 361.81   | 50.25        | 412.06     | 3.52          | 10.47   | 3.7549  | 2.74 |
> | 80                | 432.16   | 55.28        | 487.44     | 3.50          | 10.70   | 3.7858  | 2.73 |
> | 100               | 497.49   | 65.33        | 562.81     | 3.49          | 10.71   | 3.8242  | 2.74 |
> | Offline           | 1542.71  | 211.06       | 1753.77    | 3.47          | 10.82   | 3.9296  | 2.73 |
>
> >  **Note**: The latency of LLaMA-Omni has slightly changed compared to the submitted version due to variations in machine performance. To ensure fairness, we reevaluated the latency of all systems under the same conditions.
>
> From the experimental results, we observe the following:
>
> 1. Both the cascade systems and LLaMA-Omni **can achieve latencies below 320ms** (the average audio latency of GPT-4o) and allow latency control by adjusting hyperparameters. **Across all latency conditions, LLaMA-Omni consistently achieves higher ChatGPT Scores compared to the cascade baseline systems**.
>
> 2. In the offline scenario, the cascade systems achieve lower ASR-WER than LLaMA-Omni, demonstrating the superior performance of industrial TTS models. However, **in the streaming scenario, as latency decreases, we observe a significant performance drop in the cascade systems compared to the offline scenario** (evidenced by a decrease in ChatGPT Score and an increase in ASR-WER). In contrast, **LLaMA-Omni shows minimal changes in ChatGPT Score and ASR-WER as latency varies**, and even exhibits performance improvements in some cases.
> 3. In terms of speech quality, all systems experience a decline in quality as latency decreases, primarily due to an increase in discontinuities within the speech. LLaMA-Omni achieves speech quality comparable to that of SALMONN+Orca.
> 4. From the WPS metric, **LLaMA-Omni's speech rate remains almost unaffected by latency changes**. In contrast, **the overall speech rate of the cascade systems decreases significantly under low-latency conditions**, primarily due to increased pauses between word chunks, which significantly affects the overall rhythm and prosody of the speech. For example, at the lowest latency ($\Theta=1$), the overall speech rate drops to less than 60% of the offline scenario. This indicates that cascade systems face significant challenges under extremely low-latency conditions.

---

> ### Author Response · Authors · 2024-11-23
> **Author Response (4/n)**
>
> For the above results, **we have included line charts in the revised PDF illustrating the variation of each metric across different latency conditions for all models**. These charts provide a more intuitive comparison of the performance of different models. Please refer to the updated version of the paper. **We have also provided audio samples at** https://llama-omni.github.io/, allowing for a more intuitive understanding of the speech generated by different models under various latency conditions through listening.
>
> We also conducted **human evaluations** to investigate human preferences in real-time speech interaction scenarios. Specifically, we performed **side-by-side comparisons** between LLaMA-Omni ($\Omega=40$, latency=347ms) and two cascade systems: SALMONN+Orca ($\Theta=3$, latency=352ms) and Qwen2-Audio+Orca ($\Theta=3$, latency=420ms). The comparisons evaluated the systems in terms of the **helpfulness** of the responses and the **naturalness** of the generated speech, using a win/tie/lose framework. The evaluation results are as follows:
>
> | Helpfulness                     | Win  | Tie  | Lose |
> | ------------------------------- | ---- | ---- | ---- |
> | LLaMA-Omni vs. SALMONN+Orca     | 48   | 28   | 24   |
> | LLaMA-Omni vs. Qwen2-Audio+Orca | 50   | 20   | 30   |
>
> | Naturalness                     | Win  | Tie  | Lose |
> | ------------------------------- | ---- | ---- | ---- |
> | LLaMA-Omni vs. SALMONN+Orca     | 44   | 33   | 23   |
> | LLaMA-Omni vs. Qwen2-Audio+Orca | 42   | 20   | 38   |
>
> From the results, we observed that LLaMA-Omni achieved higher win rates in both helpfulness and naturalness, demonstrating that **LLaMA-Omni's responses better align with human preferences**. More details about the human evaluation can be found in Section 4.6 of the revised PDF.
>
> ---
>
> **[Q6] Decoding time.**
>
> Thank you for your suggestion. As you mentioned, comparing the total decoding time is not meaningful due to differences in the response lengths of different models. Therefore, we compared the average decoding time **per token** and **per word** for LLaMA-Omni and Qwen2-Audio+Orca in the streaming scenario. The results are shown in the table below:
>
> | Model                         | Latency (ms) | Time/token (ms) (LLM+Vocoder/TTS) | Time/word (ms) (LLM+Vocoder/TTS) |
> | ----------------------------- | ------------ | --------------------------------- | -------------------------------- |
> | LLaMA-Omni ($\Omega=20$)      | 281.41       | 29.61+14.76=44.37                 | 37.26+18.57=55.83                |
> | LLaMA-Omni ($\Omega=40$)      | 346.73       | 29.61+12.54=42.15                 | 37.26+15.78=53.04                |
> | LLaMA-Omni ($\Omega=60$)      | 412.06       | 29.61+10.71=40.32                 | 37.26+13.47=50.73                |
> | LLaMA-Omni ($\Omega=80$)      | 487.44       | 29.61+9.94=39.55                  | 37.26+12.50=49.76                |
> | Qwen2-Audio+Orca ($\Theta=3$) | 419.85       | 44.76+23.54=68.30                 | 50.49+26.56=77.05                |
> | Qwen2-Audio+Orca ($\Theta=5$) | 509.19       | 44.76+21.28=66.04                 | 50.49+24.01=74.50                |
> | Qwen2-Audio+Orca ($\Theta=7$) | 621.17       | 44.76+18.89=63.65                 | 50.49+21.31=71.80                |
> | Qwen2-Audio+Orca ($\Theta=9$) | 756.57       | 44.76+17.42=62.18                 | 50.49+19.65=70.14                |
>
> Since decoding time consists of two components: the first is **the decoding time of LLMs** (LLaMA-Omni/Qwen2-Audio), which remains constant regardless of latency, and the second is **the time required by the streaming vocoder or TTS, which varies depending on the unit/word chunk size**. To provide a clearer understanding of the time allocation, we present the two components separately in the above table. As shown, **although LLaMA-Omni’s LLM decoding involves both text and unit decoding, its average decoding time remains lower than the average time required by Qwen2-Audio for text decoding alone**. This is attributed to the parallel generation capability of LLaMA-Omni's non-autoregressive speech decoder. When comparing the streaming vocoder to streaming TTS (Orca), we observe that the average decoding time for the vocoder is consistently lower across all latency conditions. These findings demonstrate the superior decoding efficiency of LLaMA-Omni.

---

> ### Author Response · Authors · 2024-11-23
> **Author Response (5/5)**
>
> **[Q7] Multiturn conversations.**
>
> Thank you for your suggestion. While we have currently focused only on the model's performance in single-turn conversations, extending the current model to multi-turn dialogues is straightforward and natural. This can be achieved by constructing corresponding multi-turn dialogue datasets and training the model in a similar manner. We plan to explore this direction in the future.
>
> ---
>
> **[Q8] The contribution of this article.**
>
> We believe our contributions can be summarized as follows:
>
> 1. **Model Architecture**: We proposed an end-to-end model architecture capable of streaming speech response generation, achieving high-quality responses with low latency. Comparisons with prior speech language models such as SpeechGPT and cascaded systems based on streaming TTS validate the effectiveness of our proposed model.
> 2. **Data Construction**: We introduced a series of steps to adapt text instruction data to the speech domain, which are essential for speech interaction scenarios. Through extensive experiments and human evaluations, we demonstrated the effectiveness of this approach. In addition, we validated that our model exhibits robustness to voice variations when exposed to multi-speaker data, enabling it to handle real-world applications effectively.

---

> ### Comment · Reviewer_DXz7 · 2024-11-24
> **Thank you for the detailed response.**
>
> I'm very grateful for the detailed reply.
>
> Q1. Comparison with autoregressive TTS models
>
> The author point out that some traditional TTS models, such as VALL-E, need to wait for all the text before starting to generate speech. However, there are streaming TTS modules, such as Moshi and Mini-Omini used. I'm more curious about what advantages CTC-based NAT modules have over AT modules.
>
> Q2. It is not evident what benefits the end-to-end model offers over a straightforward cascaded solution where the text response is segmented into chunks and then fed into an offline TTS model.
>
> Thanks to the author's response, and this concern has been resolved.
>
> Q3. Lacking speaker generalization.
>
> Thanks to the author's response, and this concern has been resolved.
>
> Q4. Evaluation on human recorded speech.
>
> I don't agree with what the author stated that the current model has no way to evaluate ASR tasks. First of all, if the speech and text are well aligned, then it should be able to generalize the ability of dealing with text instructions + speech input. Secondly, the author can concatenate a speech instruction in front of the test speech segment to guide the model to perform the ASR task. I think the performance on ASR tasks can more intuitively reflect the degree of semantic understanding.
>
> Q5. Unconvincing evaluation metrics and results analysis.
>
> Thanks to the author's hard work. There are many additional experiments in the new version compared to the initial version. However, there are two things I don't understand. First, based on Table 2, why does the S2TIF score of LLAMA-Omini be significantly better than that of Qwen2-Audio in the offline evaluation? We all know that GPTScore has a strong length bias and should prefer longer responses. Secondly, UTMOS is an evaluation metric for TTS models and is generally applicable to evaluations when the text content is the same. For different models, the text responses are different, and this metric may not be accurate. At least for me, I think the offline speech quality of SALMONN + Orca is better than LLAMA-Omini.
>
> Q6. Decoding time.
>
> Thanks to the author's reply. WPT is a more convincing metric.
>
> Q7. Multiturn conversations.
>
> Thanks to the author's reply. Looking forward to future work.
>
> Q8. The contribution of this article.
>
> Thanks for the clarification.
>
> In conclusion, I have increased the soundness score to 3 and increased the OA to 6.

---

> ### Author Response · Authors · 2024-11-24
> **Thank you for your response! Follow-up reply below (1/2)**
>
> Thank you for your quick and detailed response! We are delighted to hear that some of the concerns have already been resolved and are very grateful that you raised the score. Next, we would be happy to continue addressing some of your questions:
>
> ---
>
> **[Q1] Comparison with streaming TTS modules used in Moshi and Mini-Omni.**
>
> Thank you for your question. The working principle of Moshi and Mini-Omni involves adding additional heads or a local Transformer after the LLM to generate corresponding speech tokens. Therefore, the model operates by **generating one text token and one speech token at each time step** (for hierarchical speech encoding, it generates all layer tokens for the frame).
>
> **Ideally, we aim for the text and speech generation processes to be approximately synchronized, meaning that as the text is generated, the corresponding speech is also produced.** This synchronization facilitates better alignment between the generated speech and text. However, one key factor is that **speech sequences are typically much longer than text sequences**. For example, let's assume the text length is 10 and the speech length is 50. In this case, Mini-Omni generates the text and the first 10 speech tokens in the first 10 time steps. After that, it still needs to autoregressively generate the remaining 40 speech tokens. During this process, the condition on the text response gradually weakens, which may result in the final speech response misaligning with the text response.
>
> Moshi addresses this issue by adding padding (PAD) to the text during training (aligning text and speech using Whisper to construct text-speech alignment information). By inserting PADs between words, it ensures that the number of text tokens corresponds more closely to the number of speech tokens per word. This enables synchronous generation of text and speech during decoding. However, this approach has two potential drawbacks:
>
> 1. During training, extracting text-speech alignment information to construct the dataset increases the complexity of data preprocessing.
> 2. Forcing the LLM to generate text responses with PADs disrupts its natural text generation mode, which might harm its inherent text generation capability.
>
> In contrast, our approach first upsamples the LLM’s output states to ensure even alignment between text and speech. Then, the model **uses CTC to adaptively learn the alignment relationships, naturally supporting synchronized generation of text and speech**. We believe this offers an advantage over the aforementioned autoregressive models.

---

> > ### Author Response · Authors · 2024-11-24
> > **Thank you for your response! Follow-up reply below (2/2)**
> >
> > **[Q2] ASR evaluation.**
> >
> > Thank you for your valuable suggestions! We quickly experimented with the two methods you mentioned. Specifically, we synthesized a speech instruction with the content: "Please transcribe the following speech into text," and appended it to the test speech (with a 500ms pause in between). Additionally, we modified the text instruction to: "Transcribe the speech into text, ensuring your response includes only the content of the speech without any additional information." We then tested this setup on the LibriSpeech `test-clean` dataset.
> >
> > Our findings show that, in most cases, the model successfully performs the ASR task. However, occasionally, the model outputs extra information (despite our instruction explicitly asking it not to), such as "The speaker says..." or "Here is the transcription of the speech:". In a few cases, the model still treated the speech as a question and generated a response instead of completing the ASR task. These erroneous outputs result in a relatively high WER of 50.19%. Examples of outputs are as follows:
> >
> > > Case1-pred: The dull light faded more faintly above the page where another equation began to unfold itself slowly and spread its widening tail.
> > >
> > > Case1-answer: the dull light fell more faintly upon the page whereon another equation began to unfold itself slowly and to spread abroad its widening tail
> > >
> > > Case2-pred: Here is the transcription of the speech: Well now, I declare, I have a head and so has my stick.
> > >
> > > Case2-answer: well now ennis i declare you have a head and so has my stick
> > >
> > > Case3-pred: The speaker is describing a quiet and peaceful atmosphere in a schoolroom, where the only sounds are the dust settling and a bell ringing.
> > >
> > > Case3-answer: but the dusk deepening in the schoolroom covered over his thoughts the bell rang
> >
> > We believe the primary reason lies in the model not having encountered this type of task during training. Additionally, the training instructions were overly simplistic and uniform (e.g., *"Please answer the questions in the user’s input speech"*), which limited the model's ability to generalize to ASR tasks.
> >
> > The simplest way to address this issue is to incorporate a small amount of ASR-specific instruction data into the training process, as demonstrated in our previous response. In the future, we also plan to explore incorporating more diverse text instruction data during training to enhance the model's ability to generalize across different tasks.
> >
> >
> >
> > ---
> >
> > **[Q3] Why does the S2TIF score of LLAMA-Omini be significantly better than that of Qwen2-Audio in the offline evaluation?**
> >
> > Thank you for your question. We believe this phenomenon is primarily due to two reasons:
> >
> > 1. **Model Foundations**: Qwen2-Audio is built upon the Qwen-7B model, while LLaMA-Omni is based on the Llama-3.1-8B-Instruct model. Therefore, **it is reasonable that LLaMA-Omni exhibits superior reasoning capabilities compared to Qwen2-Audio**. We observed that Qwen2-Audio occasionally fails to follow instructions or produces clearly erroneous outputs, which could contribute to its lower scores.
> > 2. **Evaluation Prompt Context**: In our evaluation prompt, we informed GPT-4o that the evaluation is set in a speech interaction scenario. **We speculate that GPT-4o's evaluation preferences may vary depending on the scenario**, potentially leading to a reduced preference for longer responses in this specific context.
> >
> > ---
> >
> > **[Q4] UTMOS metric.**
> >
> > Thank you for your question. We agree that the UTMOS evaluation metric may have some limitations and may not fully capture human preferences. Honestly, in offline scenarios, we also believe Orca’s speech synthesis performance is superior to LLaMA-Omni’s, given that Orca is an industrial TTS model.
> >
> > **We speculate that one reason for Orca’s slightly lower UTMOS score compared to LLaMA-Omni could be the impact of speech length on the scoring.** Longer speech samples might contain more errors, which are more likely to be detected by the model. One piece of supporting evidence for this is that Qwen2-Audio + Orca generally has lower UTMOS scores than SALMONN + Orca, even though both use the same TTS model. The key difference is that Qwen2-Audio responses tend to be longer than SALMONN’s. Therefore, the longer response length of SALMONN compared to LLaMA-Omni may have contributed to the phenomenon mentioned above.
> >
> > Nevertheless, we believe that UTMOS can still serve as a useful reference for speech quality in automatic evaluations. At the same time, we plan to conduct human evaluations in offline scenarios in the future for a more comprehensive and accurate assessment.

---

### Official Review · Reviewer_ppgU · 2024-10-29

**Soundness:** 3
**Presentation:** 2
**Contribution:** 2
**Rating:** 6
**Confidence:** 4

**Summary:**

This paper proposes LLaMA-Omni, an end-to-end model architecture designed for low-latency, high-quality speech interaction with large language models (LLMs). LLaMA-Omni integrates multiple components: a pre-trained speech encoder, a speech adaptor, a large language model, and a streaming speech decoder. To enable more natural, human-like speech interactions, the authors introduce InstructS2S-200K, a dataset of 200K speech instructions and responses crafted in a conversational, oral style, to fine-tune the model for speech-based dialogue. Experimental results show that LLaMA-Omni achieves low-latency speech responses while reducing training costs compared to similar models.

**Strengths:**

1. **Conversational Data with InstructS2S-200K:** The proposed InstructS2S-200K dataset addresses a notable gap in conversational, oral-style speech instruction data. By providing a large, purpose-built dataset tailored to natural, interactive speech patterns, this work supports more effective alignment of LLM responses with human conversational norms.
2. **Cost-Effective Training Process:** The two-stage training process adopted in LLaMA-Omni demonstrates a practical approach to reducing training costs. Compared to prior end-to-end training methods, this staged training setup optimizes resource use, enabling faster model development without compromising performance.

**Weaknesses:**

1. **Lack of Novelty:** While the authors suggest that they proposed a "novel model architecture," it appears to primarily build on a combination of existing methods without a clear breakthrough in architecture:
   1) The connection of a speech encoder to an LLM via a speech adaptor has been widely explored in prior research.
   2) The streaming TTS module seems to be directly based on previous designs, specifically following the approach outlined in [1].

    In summary, while LLaMA-Omni successfully assembles a functional pipeline for speech-based interactions with LLMs, the architecture seems to build on prior work without delivering significant innovation in model design, which somewhat limits the paper’s impact from a research innovation standpoint. Further discussion is needed to emphasize the unique contribution of this work.
2. **Questionable Experiment Results:** Some experiment results in this paper is questionable:
   1) **ChatGPT-Score in Table 2:** While LLaMA-Omni shows improved performance over baseline models in ChatGPT-Score, this result is potentially misleading. The evaluation prompt used (detailed in Appendix A) was crafted to reward concise responses and penalize extraneous information. This prompt design aligns closely with the prompts used to construct the InstructS2S-200K dataset, creating a circular evaluation condition where the model is likely to excel by design. A more objective and meaningful approach would involve conducting a human evaluation to capture human preferences more accurately.
   2) **WER & CER in Table 2:** In Table 2, Qwen2-Audio + TTS shows an unexpectedly high WER of 55.72%, yet it achieves a relatively moderate S2SIF score of 2.32/2.58, which implies a more acceptable response quality. This discrepancy suggests an inconsistency, as such a high WER would typically indicate a low intelligibility level that should correlate with a poor S2SIF score. This contradiction casts doubt on the reliability of the WER measurements or, potentially, the effectiveness of the S2SIF scoring methodology when used alongside TTS outputs.

        A likely explanation for these irregularities lies in the TTS model used, VITS, which is primarily trained on **short audio clips** [2]. This limitation may hinder VITS’s ability to process longer inputs effectively, causing high error rates when it encounters the longer outputs of models like Qwen2-Audio and SALMONN. Sending unchunked, lengthy responses directly to VITS without segmentation could result in degraded performance, as evidenced by the elevated WER and CER scores.

        To address these limitations and obtain more accurate metrics, it would be more appropriate to adopt a TTS model trained for longer utterances or to chunk input text before passing it to VITS. Such adjustments would likely yield more realistic WER and CER values and provide a fairer basis for comparing baseline models.
    3) **Average decoding time in Table 4:** The decoding time reported in Table 4 does not convincingly demonstrate the superiority of the proposed streaming decoding architecture. The average decoding time is measured for generating complete speech responses; however, since baseline methods produce longer responses, they naturally require more time for decoding. This dependency on response length makes the reported decoding times trivial and limits their effectiveness in showcasing any advantages of the proposed approach. To provide a fair and meaningful comparison, it would be more appropriate to compare with Qwen2-Audio + streaming TTS model and report decoding time per token rather than total decoding time.
3. **Limited speaker support:** While InstructS2S-200K provides valuable conversational data, it has limitations due to the lack of speaker diversity. The input speech in this dataset is synthesized using CosyVoice-300M-SFT, which represents a single voice and does not capture the variability in accents, pitch, tone, and other speaker characteristics found in real-world interactions. This limitation restricts LLaMA-Omni’s adaptability to diverse user-profiles and reduces its effectiveness in applications requiring personalized or varied speaker responses. Integrating a broader range of voices could significantly improve LLaMA-Omni’s applicability in real-world scenarios.


[1] Fang Q, Ma Z, Zhou Y, et al. CTC-based Non-autoregressive Textless Speech-to-Speech Translation[J]. arXiv preprint arXiv:2406.07330, 2024.

[2] Kim J, Kong J, Son J. Conditional variational autoencoder with adversarial learning for end-to-end text-to-speech[C]//International Conference on Machine Learning. PMLR, 2021: 5530-5540.

**Questions:**

See above.

---

> ### Author Response · Authors · 2024-11-23
> **Author Response (1/n)**
>
> Thank you for providing your valuable and constructive feedback! Please find below the responses to each comment.
>
> ---
>
> **[Q1] Lack of novelty.**
>
> Thank you for your question. Our speech decoder draws inspiration from prior works that use CTC for speech generation, such as CTC-S2UT [1], NAST-S2x [2], and StreamSpeech [3]. Similar to [2] and [3], we also adopt a CTC-based model for streaming discrete unit generation. However, our work still differs from theirs in several key aspects.
>
> 1. **All previous models utilize an encoder-decoder architecture, whereas we are the first to adopt a decoder-only architecture for CTC-based streaming unit generation**. This design is more consistent with the architecture of large language models and is potentially easier to scale up.
> 2. Prior works have only validated the feasibility of CTC-based models on sentence-level machine translation tasks (average unit sequence length on CVSS Fr-En: 101.3). In contrast, **we are the first to demonstrate the feasibility of using a CTC-based non-autoregressive model for generating long-sequence discrete units** (average unit sequence length: 553.6).
>
> Besides, to the best of our knowledge, we are the first to successfully integrate this architecture with LLMs and validate its effectiveness. Compared to prior speech-language models, such as SpeechGPT, our framework achieves **lower training costs** and **higher quality**. Therefore, while some submodules in our work resemble those in earlier studies, **we believe that proposing this framework and demonstrating its effectiveness also represents a meaningful contribution**.
>
> ---
>
> **[Q2] Questionable Experiment Results.**
>
> We sincerely thank you for pointing out potential issues in our experimental results. Based on your suggestions, we conducted a comprehensive review of our experiments and identified several key issues, which we have thoroughly corrected:
>
> 1. **ASR-WER/CER**: Regarding your observation of unusually high ASR-WER/CER scores of the Qwen2-Audio+TTS Model, we found this was due to an oversight. When calculating ASR-WER/CER, the Whisper model incorrectly truncated the speech to 30 seconds, resulting in the loss of content after the 30-second mark, which led to incorrect ASR-WER/CER scores. **We have corrected this issue and conducted a complete re-evaluation.**
> 2. **ChatGPT Score**: The evaluation prompt we previously used may have introduced bias. **We have removed all potentially bias-inducing priors from the prompt** and simply instructed the model to give a single score based on helpfulness, relevance, fluency, and suitability for speech interaction, without providing any explanations that might introduce bias (details in Appendix A of the revised PDF). **We also conducted human evaluations following your suggestions**, and the results are presented below.
> 3. **TTS Model**: Apologies for the previous unclear explanation, which may have caused misunderstandings. **The VITS model we used is integrated within the [Coqui TTS framework](https://github.com/coqui-ai/TTS), which performs text segmentation before speech synthesis to ensure accurate synthesis even for long texts**. In fact, the speech responses in our training dataset were also synthesized using this model. However, following your suggestion and to achieve a more comprehensive and meaningful comparison, especially in the streaming scenario, **we have replaced the cascaded baseline systems with Qwen2-Audio/SALMONN + [Orca](https://github.com/Picovoice/orca), where Orca is an industrial TTS model that supports both streaming and offline speech synthesis**. When using Orca for streaming speech synthesis, we need to set a **word chunk size** $\Theta$, which means that speech synthesis is triggered every time $\Theta$ new words arrive. In our experiments, we varies $\Theta$ within the range of $[1, 3, 5, 7, 9]$ to control the response latency of cascaded systems.

---

> ### Author Response · Authors · 2024-11-23
> **Author Response (2/n)**
>
> **We conducted a comprehensive re-evaluation of all models in both offline and streaming scenarios**. Specifically, we evaluate the model using the following metrics:
>
> 1. **ChatGPT Score [Updated]**: We use GPT-4o to rate the model's responses. Following the suggestions from reviewers ppgU and DXz7, we have revised the evaluation prompt (details in Appendix A of the revised PDF) to remove potential prior biases.
> 2. **ASR-WER [Unchanged]**: This metric evaluates the alignment between text and speech responses. We have corrected the errors that occurred during the previous ASR process.
> 3. **UTMOS [Unchanged]**: This metric assesses the quality of the generated speech.
> 4. **WPS (Words Per Second) [New]**: This metric measures the speaking rate of the generated speech, which is calculated as the average number of words per second in the generated speech.
> 5. **Latency [Updated]**: Latency refers to the **total delay** between the user's speech input and the moment they hear the speech response. This can be further divided into the **decoding latency of the LLM** and **the latency introduced by the Vocoder or TTS model**.
>
> The numerical results under different latency conditions for the cascade systems (SALMONN + Orca, Qwen2-Audio + Orca) and LLaMA-Omni are as follows:
>
> **[SALMONN + Orca]**
>
> | $\mathbf{\Theta}$ | Latency-LLM (ms) | Latency-TTS (ms) | Latency-Total (ms) | ChatGPT Score | ASR-WER | UTMOS  | WPS  |
> | ----------------- | ---------------- | ---------------- | ------------------ | ------------- | ------- | ------ | ---- |
> | 1                 | 212.45           | 19.71            | 232.16             | 3.28          | 6.64    | 3.0947 | 1.86 |
> | 3                 | 316.59           | 35.08            | 351.67             | 3.09          | 8.65    | 3.7338 | 2.91 |
> | 5                 | 428.79           | 32.08            | 460.87             | 3.04          | 9.23    | 3.7750 | 3.18 |
> | 7                 | 536.47           | 45.90            | 582.37             | 3.06          | 8.49    | 3.7972 | 3.20 |
> | 9                 | 659.57           | 69.38            | 728.95             | 3.23          | 7.07    | 3.8060 | 3.22 |
> | Offline           | 4274.48          | 1049.59          | 5324.07            | 3.40          | 3.78    | 3.8286 | 3.31 |
>
> **[Qwen2-Audio + Orca]**
>
> | $\mathbf{\Theta}$ | Latency-LLM (ms) | Latency-TTS (ms) | Latency-Total (ms) | ChatGPT Score | ASR-WER | UTMOS  | WPS  |
> | ----------------- | ---------------- | ---------------- | ------------------ | ------------- | ------- | ------ | ---- |
> | 1                 | 289.46           | 19.15            | 308.61             | 2.79          | 25.25   | 2.8597 | 1.90 |
> | 3                 | 381.66           | 38.19            | 419.85             | 2.95          | 13.30   | 3.5529 | 2.91 |
> | 5                 | 470.55           | 38.64            | 509.19             | 2.93          | 13.00   | 3.5865 | 3.08 |
> | 7                 | 568.22           | 52.95            | 621.17             | 3.07          | 10.59   | 3.5739 | 3.11 |
> | 9                 | 675.55           | 81.02            | 756.57             | 3.14          | 9.81    | 3.6016 | 3.13 |
> | Offline           | 7062.93          | 2361.49          | 9424.42            | 3.38          | 6.77    | 3.6119 | 3.30 |
>
> **[LLaMA-Omni]**
>
> | $\mathbf{\Omega}$ | Latency-LLM (ms) | Latency-Vocoder (ms) | Latency-Total (ms) | ChatGPT Score | ASR-WER | UTMOS  | WPS  |
> | ----------------- | ---------------- | -------------------- | ------------------ | ------------- | ------- | ------ | ---- |
> | 10                | 206.03           | 30.15                | 236.18             | 3.54          | 9.84    | 3.2304 | 2.76 |
> | 20                | 236.18           | 45.23                | 281.41             | 3.56          | 9.91    | 3.4748 | 2.75 |
> | 40                | 301.51           | 45.23                | 346.73             | 3.52          | 10.37   | 3.6688 | 2.74 |
> | 60                | 361.81           | 50.25                | 412.06             | 3.52          | 10.47   | 3.7549 | 2.74 |
> | 80                | 432.16           | 55.28                | 487.44             | 3.50          | 10.70   | 3.7858 | 2.73 |
> | 100               | 497.49           | 65.33                | 562.81             | 3.49          | 10.71   | 3.8242 | 2.74 |
> | Offline           | 1542.71          | 211.06               | 1753.77            | 3.47          | 10.82   | 3.9296 | 2.73 |
>
> >  **Note**: The latency of LLaMA-Omni has slightly changed compared to the submitted version due to variations in machine performance. To ensure fairness, we reevaluated the latency of all systems under the same conditions.

---

> ### Author Response · Authors · 2024-11-23
> **Author Response (3/n)**
>
> From the experimental results, we observe the following:
>
> 1. Both the cascade systems and LLaMA-Omni **can achieve latencies below 320ms** (the average audio latency of GPT-4o) and allow latency control by adjusting hyperparameters. **Across all latency conditions, LLaMA-Omni consistently achieves higher ChatGPT Scores compared to the cascade baseline systems**.
>
> 2. In the offline scenario, the cascade systems achieve lower ASR-WER than LLaMA-Omni, demonstrating the superior performance of industrial TTS models. However, **in the streaming scenario, as latency decreases, we observe a significant performance drop in the cascade systems compared to the offline scenario** (evidenced by a decrease in ChatGPT Score and an increase in ASR-WER). In contrast, **LLaMA-Omni shows minimal changes in ChatGPT Score and ASR-WER as latency varies**, and even exhibits performance improvements in some cases.
> 3. In terms of speech quality, all systems experience a decline in quality as latency decreases, primarily due to an increase in discontinuities within the speech. LLaMA-Omni achieves speech quality comparable to that of SALMONN+Orca.
> 4. From the WPS metric, **LLaMA-Omni's speech rate remains almost unaffected by latency changes**. In contrast, **the overall speech rate of the cascade systems decreases significantly under low-latency conditions**, primarily due to increased pauses between word chunks, which significantly affects the overall rhythm and prosody of the speech. For example, at the lowest latency ($\Theta=1$), the overall speech rate drops to less than 60% of the offline scenario. This indicates that cascade systems face significant challenges under extremely low-latency conditions.
>
> For the above results, **we have included line charts in the revised PDF illustrating the variation of each metric across different latency conditions for all models**. These charts provide a more intuitive comparison of the performance of different models. Please refer to the updated version of the paper. **We have also provided audio samples at** https://llama-omni.github.io/, allowing for a more intuitive understanding of the speech generated by different models under various latency conditions through listening.
>
> We also conducted **human evaluations** to investigate human preferences in real-time speech interaction scenarios. Specifically, we performed **side-by-side comparisons** between LLaMA-Omni ($\Omega=40$, latency=347ms) and two cascade systems: SALMONN+Orca ($\Theta=3$, latency=352ms) and Qwen2-Audio+Orca ($\Theta=3$, latency=420ms). The comparisons evaluated the systems in terms of the **helpfulness** of the responses and the **naturalness** of the generated speech, using a win/tie/lose framework. The evaluation results are as follows:
>
> | Helpfulness                     | Win  | Tie  | Lose |
> | ------------------------------- | ---- | ---- | ---- |
> | LLaMA-Omni vs. SALMONN+Orca     | 48   | 28   | 24   |
> | LLaMA-Omni vs. Qwen2-Audio+Orca | 50   | 20   | 30   |
>
> | Naturalness                     | Win  | Tie  | Lose |
> | ------------------------------- | ---- | ---- | ---- |
> | LLaMA-Omni vs. SALMONN+Orca     | 44   | 33   | 23   |
> | LLaMA-Omni vs. Qwen2-Audio+Orca | 42   | 20   | 38   |
>
> From the results, we observed that LLaMA-Omni achieved higher win rates in both helpfulness and naturalness, demonstrating that **LLaMA-Omni's responses better align with human preferences**. More details about the human evaluation can be found in Section 4.6 of the revised PDF.

---

> ### Author Response · Authors · 2024-11-23
> **Author Response (4/4)**
>
> **[Q3] Average decoding time.**
>
> Thank you for your suggestion. As you mentioned, comparing the total decoding time is not meaningful due to differences in the response lengths of different models. Following your advice, we compared the average decoding time **per token** and **per word** for LLaMA-Omni and Qwen2-Audio+Orca in the streaming scenario. The results are shown in the table below:
>
> | Model                         | Latency (ms) | Time/token (ms) (LLM+Vocoder/TTS) | Time/word (ms) (LLM+Vocoder/TTS) |
> | ----------------------------- | ------------ | --------------------------------- | -------------------------------- |
> | LLaMA-Omni ($\Omega=20$)      | 281.41       | 29.61+14.76=44.37                 | 37.26+18.57=55.83                |
> | LLaMA-Omni ($\Omega=40$)      | 346.73       | 29.61+12.54=42.15                 | 37.26+15.78=53.04                |
> | LLaMA-Omni ($\Omega=60$)      | 412.06       | 29.61+10.71=40.32                 | 37.26+13.47=50.73                |
> | LLaMA-Omni ($\Omega=80$)      | 487.44       | 29.61+9.94=39.55                  | 37.26+12.50=49.76                |
> | Qwen2-Audio+Orca ($\Theta=3$) | 419.85       | 44.76+23.54=68.30                 | 50.49+26.56=77.05                |
> | Qwen2-Audio+Orca ($\Theta=5$) | 509.19       | 44.76+21.28=66.04                 | 50.49+24.01=74.50                |
> | Qwen2-Audio+Orca ($\Theta=7$) | 621.17       | 44.76+18.89=63.65                 | 50.49+21.31=71.80                |
> | Qwen2-Audio+Orca ($\Theta=9$) | 756.57       | 44.76+17.42=62.18                 | 50.49+19.65=70.14                |
>
> Since decoding time consists of two components: the first is **the decoding time of LLMs** (LLaMA-Omni/Qwen2-Audio), which remains constant regardless of latency, and the second is **the time required by the streaming vocoder or TTS, which varies depending on the unit/word chunk size**. To provide a clearer understanding of the time allocation, we present the two components separately in the above table. As shown, **although LLaMA-Omni’s LLM decoding involves both text and unit decoding, its average decoding time remains lower than the average time required by Qwen2-Audio for text decoding alone**. This is attributed to the parallel generation capability of LLaMA-Omni's non-autoregressive speech decoder. When comparing the streaming vocoder to streaming TTS (Orca), we observe that the average decoding time for the vocoder is consistently lower across all latency conditions. These findings demonstrate the superior decoding efficiency of LLaMA-Omni.
>
> ---
>
> **[Q4] Limited speaker support.**
>
> Thank you for your suggestion. As you pointed out, the speech instructions in our training set were synthesized using CosyVoice, which includes only two standard voices (male and female). **However, since our speech encoder (whisper) remains frozen throughout training, its inherent ability to handle multi-speaker speech remains intact.**
>
> To verify this, we synthesized a new training and evaluation dataset using the [**fish-speech-1.4**](https://huggingface.co/fishaudio/fish-speech-1.4) model, with a randomly selected voice for each synthesis. **This ensured that all data samples have unique and non-overlapping voices.** We then trained a new model on this dataset while keeping all other settings unchanged. Finally, we evaluated the model on two test sets: the **single-speaker test set synthesized by CosyVoice** (the same test set used in other experiments) and the **multi-speaker test set synthesized by fish-speech-1.4**. The results are presented in the table below.
>
> | ID   | Training   | Evaluation | S2TIF ChatGPT Score |
> | ---- | ---------- | ---------- | ------------------- |
> | Exp1    | CosyVoice  | CosyVoice  | 3.99                |
> | Exp2    | CosyVoice  | FishSpeech | 4.07                |
> | Exp3    | FishSpeech | CosyVoice  | 3.96                |
> | Exp4    | FishSpeech | FishSpeech | 4.19                |
>
> Comparing Exp1 and Exp2, we observe that **even the model trained on only two standard voices performs reasonably well on the multi-speaker test set**, with some improvement observed (possibly due to differences in synthesis accuracy between CosyVoice and FishSpeech). However, comparing Exp2 and Exp4, we find that **the model trained on multi-speaker data performs better on the multi-speaker test set** (4.07 vs. 4.19). This demonstrates the benefits of incorporating voice diversity during training.
>
> In summary, we agree with your point that training with multi-speaker data can enhance the model's robustness. However, **we believe that the current model is already sufficient to handle real-world scenarios effectively**.
>
> ---
>
> **References:**
>
> [1] CTC-based Non-autoregressive Textless Speech-to-Speech Translation.
>
> [2] A Non-autoregressive Generation Framework for End-to-End Simultaneous Speech-to-Speech Translation.
>
> [3] StreamSpeech: Simultaneous Speech-to-Speech Translation with Multi-task Learning.

---

> ### Comment · Reviewer_ppgU · 2024-11-24
>
> I thank the authors for their response.
>
> While the response addressed many of my initial concerns, it also highlights a critical weakness of Llama-Omni.
>
> As shown in Table 2, there is a significant gap (0.52) between the S2TIF and S2SIF scores of Llama-Omni, primarily due to the limitations of the adopted CTC speech decoder. In contrast, Orca demonstrates a decline of less than 0.1 in its ChatGPT Score, with acceptable latency as evidenced by Table 6. This suggests that integrating Llama-Omni (Text) with Orca could potentially yield a much better ChatGPT Score without compromising latency. Compared to this straightforward alternative, Llama-Omni is held back by its CTC speech decoder, rendering the proposed architecture impractical and lacking meaningful application.
>
> Furthermore, another major contribution of the paper—the InstructS2S-200K dataset—also raises concerns. While the goal of creating responses tailored for speech interaction is commendable, the proposed method involves rewriting responses using an LLM. This raises the question: would converting the rewrite prompt into a system prompt and directly utilizing it during generation achieve similar results with the desired characteristics?

---

> > ### Author Response · Authors · 2024-11-24
> > **Thank you for your response!**
> >
> > Thank you for your quick response! We're glad to hear that many of your issues have been resolved. We will conduct experiments on LLaMA-Omni (Text) + Orca and provide the results as soon as possible.

---

> > ### Author Response · Authors · 2024-11-25
> > **Follow-up responses (1/2)**
> >
> > Thank you again for your feedback! Below are our responses to your new questions.
> >
> > ---
> >
> > **[Q1] Comparison with LLaMA-Omni (Text) + Orca.**
> >
> > Thank you for your questions. Following your suggestions, we conducted experiments with LLaMA-Omni (Text) + Orca, and the results are shown in the table below.
> >
> > **[LLaMA-Omni (Text) + Orca]**
> >
> > | $\mathbf{\Theta}$ | Latency (ms) | ChatGPT Score | ASR-WER | UTMOS  | WPS  |
> > |--------------------|----------|----------|------------|---------------|---------|
> > | 1                 | 205.99 | 3.83          | 8.68    | 3.2856 | 1.85 |
> > | 3                 | 279.69 | 3.60          | 10.98   | 3.8496 | 2.75 |
> > | 5            | 336.34 | 3.53          | 11.14   | 3.9183 | 3.02 |
> > | 7                 | 425.22 | 3.64          | 9.49    | 3.9024 | 3.04 |
> > | 9                 | 516.44 | 3.75          | 7.56    | 3.8925 | 3.04 |
> > | Offline           | / | 3.89          | 4.94    | 3.8999 | 3.07 |
> >
> > **[LLaMA-Omni (Speech)]**
> >
> > | $\mathbf{\Omega}$ | Latency (ms) | ChatGPT Score | ASR-WER | UTMOS  | WPS  |
> > | ----------------- | ------------ | ------------- | ------- | ------ | ---- |
> > | 10                | 236.18       | 3.54          | 9.84    | 3.2304 | 2.76 |
> > | 20                | 281.41       | 3.56          | 9.91    | 3.4748 | 2.75 |
> > | 40                | 346.73       | 3.52          | 10.37   | 3.6688 | 2.74 |
> > | 60                | 412.06       | 3.52          | 10.47   | 3.7549 | 2.74 |
> > | 80                | 487.44       | 3.50          | 10.70   | 3.7858 | 2.73 |
> > | 100               | 562.81       | 3.49          | 10.71   | 3.8242 | 2.74 |
> > | Offline           | /            | 3.47          | 10.82   | 3.9296 | 2.73 |
> >
> > From the comparison with the results of LLaMA-Omni (Speech), we can observe the following:
> >
> > 1. First, in offline scenarios, LLaMA-Omni (Text) + Orca achieved a higher ChatGPT Score and lower ASR-WER, which aligns with our expectations, as Orca, being an industrial TTS model, offers superior speech synthesis quality.
> >
> > 2. In low-latency streaming scenarios, for instance, comparing LLaMA-Omni (Text) + Orca ($\Theta=3$) with LLaMA-Omni (Speech) ($\Omega=20$), both systems maintain a latency of approximately 280ms. Here, the ChatGPT Scores (3.60 vs. 3.56) and ASR-WER (10.98 vs. 9.91) are relatively close, indicating that the content quality of the generated speech is comparable.
> >
> > 3. As for speech quality, although the UTMOS metric suggests that the cascaded system using Orca achieves higher speech quality, our previous human evaluations indicate that under low-latency conditions, human evaluators have a stronger preference for LLaMA-Omni's speech. This is because systems based on Orca often introduce pauses between word chunks, resulting in unnatural speech rhythm and prosody. This is also reflected in the WPS (Words Per Second) metric, where a decrease in speech rate can be observed (3.07→2.75) when $\Theta=3$.
> >
> > **In summary, while the cascaded system can outperform LLaMA-Omni under certain conditions, we believe our proposed model is also contributive, particularly in low-latency streaming scenarios**. Additionally, we believe that end-to-end models have a higher potential ceiling compared to cascaded systems, as they can directly condition on input speech during the generation of speech responses, rather than relying solely on text-based responses. This allows for the possibility of generating responses that adapt to paralinguistic cues in the input speech, although we have not yet explored this aspect.
> >
> > Compared to previous end-to-end models like SpeechGPT, our model demonstrates clear advantages in multiple dimensions. We believe that our work will inspire further research into end-to-end speech-language models and ultimately surpass cascaded systems in the future.

---

> > > ### Author Response · Authors · 2024-11-25
> > > **Follow-up responses (2/2)**
> > >
> > > **[Q2] Would converting the rewrite prompt into a system prompt and directly utilizing it during generation achieve similar results with the desired characteristics?**
> > >
> > > Thank you for your question. We agree that controlling the model's generation style through prompts during inference is a viable approach. However, we believe that performing response rewriting in advance offers the following advantages:
> > >
> > > 1. Controlling the output style through prompts during inference relies on the 8B-scale model itself to manage style. On the other hand, rewriting responses during the data preparation phase allows us to leverage larger-scale LLMs (e.g., we used Llama3-70B-Instruct). In our early experiments with different LLMs during data preparation, we observed that larger models handle output styles across various scenarios more effectively. This approach essentially distills part of the 70B model's capabilities into the 8B model, which we believe leads to better overall performance.
> > >
> > > 2. If text responses are not rewritten and are directly synthesized into speech, several issues may arise. Text responses may contain symbols or elements that the TTS system cannot properly synthesize, leading to errors in the final speech output. Additionally, longer text responses result in longer synthesized speech, increasing the difficulty of model training. Furthermore, training the model on unaltered text responses and their corresponding speech data while prompting it during inference to generate a different style of text response can create a training-inference mismatch, potentially degrading the quality of the final synthesized speech.
> > >
> > > In summary, we believe that rewriting responses in advance for data preparation offers several advantages. Moreover, in speech interaction scenarios, tailoring data specifically for the task is a reasonable approach, as supported by findings in previous studies [1].
> > >
> > > **References:**
> > >
> > > [1] Cho et al. Speechworthy instruction-tuned language models. EMNLP 2024.

---

> > > > ### Author Response · Authors · 2024-11-26
> > > > **Looking forward to your feedback!**
> > > >
> > > > Dear Reviewer ppgU,
> > > >
> > > > Thank you again for your valuable comments! We have provided new responses to your follow-up questions. We would like to know if our replies have addressed your concerns, and if you have any further questions. Looking forward to your feedback.
> > > >
> > > > Best,
> > > >
> > > > Authors

---

> > > > > ### Author Response · Authors · 2024-11-28
> > > > > **Kind Reminder**
> > > > >
> > > > > Dear Reviewer ppgU,
> > > > >
> > > > > Thank you again for the time and effort you’ve dedicated to reviewing our work. As the discussion phase is coming to a close, we kindly ask if you could review our follow-up responses. **We’re grateful that you’ve mentioned most of your initial concerns have been addressed, and we have also provided replies to your follow-up questions.**
> > > > >
> > > > > **If there are no further concerns, we would be grateful if you could reconsider the score.**
> > > > >
> > > > > Thank you for your time.
> > > > >
> > > > > Best regards,
> > > > >
> > > > > Authors

---

> ### Comment · Reviewer_ppgU · 2024-11-29
>
> Apologies for the delayed response—I got sick for the past few days.
>
> > Regarding the archtecture
>
> After reviewing the new experimental results, I believe they support my concerns that the current architecture generally underperforms compared to the more straightforward pipeline approach. This significantly weakens the overall contribution of the architecture.
>
> > Regarding the InstructS2S-200K
>
> I reviewed the paper and noticed that the authors directly prompt the 70B model to generate responses without any filtering. This could lead to undesirable behaviors compared to the preference data provided in [1]. While the authors claim they can distill knowledge from the 70B model to the 8B model, I believe this ‘advantage’ may diminish as the size of the text-LLM scales. For instance, if the backbone LLM were switched to Llama-3-70B, directly using the same prompting approach would likely produce the same results.
>
> Besides, fine-tuning LLMs with data generated by another model may lead to catastrophic forgetting of their ability. While the author conduct experiments in response to Reviewer 4A7w's concern, MMLU is not reliable for evaluating forgetting [2]. A more percise evaluation should be conduct on instruction following benchmarks (eg. AlpacaEval). In contrast, the prompting approach does not disturb the model parameter. This limits the applicability of this dataset on other Speech-LLMs.
>
> > Conclusion.
>
> Overall, I think this paper is more engineering and does not bring refreshing new insights. At the same time, its performance does not have advantage compared to the straight forward pipeline approach. Therefore, I think I don't want to push for acceptance and tend to maintain my current score.
>
> BTW, although I acknowledge the efforts in the rebuttal of the authors, I don't think it is a good idea to submit an unfinished manuscript and redo almost all the experiment and analysis during the review stage.
>
> [1] Cho et al. Speechworthy instruction-tuned language models. EMNLP 2024.
>
> [2] Yang, Zhaorui, et al. "Self-distillation bridges distribution gap in language model fine-tuning." arXiv preprint arXiv:2402.13669 (2024).

---

> > ### Author Response · Authors · 2024-11-29
> > **Thank you for your response! Please find our further responses below (1/2).**
> >
> > Dear Reviewer ppgU,
> >
> > Thank you once again for reviewing our response and providing detailed feedback. We are sorry to hear that you’ve been unwell, and we sincerely hope you are feeling better now. After reading your feedback, we would like to provide some further clarifications.
> >
> > > Regarding the archtecture
> >
> > Regarding the comparison between LLaMA-Omni (Speech) and LLaMA-Omni (Text) + Orca, **we disagree with the statement "the current architecture generally underperforms compared to the more straightforward pipeline approach."** A simple comparison based solely on the ChatGPT Score is not comprehensive, as it is based on transcribed text and can only evaluate the quality of the content. For example, in a low-latency comparison, Cascade ($\Theta=1$) vs. LLaMA-Omni ($\Omega=10$), the cascaded system does have a higher ChatGPT Score (3.83 vs. 3.54), but the generated speech rhythm is very unnatural, with excessive pauses (WPS drops by 40%). When comparing Cascade ($\Theta=3$) vs. LLaMA-Omni ($\Omega=20$), the ChatGPT Score shows no significant difference (3.60 vs. 3.56), but the cascade system's speech still exhibits unnatural rhythm. You can listen to the audio generated by Qwen2/SALMONN + Orca at low latency on https://llama-omni.github.io/ to understand what we mean by "unnatural rhythm," as there are noticeable pauses between word chunks. **At the same time, we would like to emphasize that we have conducted human evaluations (as mentioned in our previous response), which demonstrate that the speech generated by LLaMA-Omni at low latency is more in line with human preferences compared to the cascaded system. We believe this provides more compelling evidence than automated evaluation metrics.**
> >
> > Stepping away from specific numerical comparisons in this context, **we understand that there are different perspectives on the comparison between end-to-end modeling and cascade systems**. Cascade systems, due to the mature components in their pipeline, are generally easier to achieve good results with a lower cost. In contrast, end-to-end modeling faces numerous challenges, such as data scarcity and model design, making it difficult to immediately reach or surpass the performance of cascade systems. **However, this does not mean that exploring end-to-end models is without merit.** For example, early speech translation systems were almost based on cascaded ASR and MT components. Starting in 2016, researchers began proposing end-to-end speech translation models [1]. In the years that followed, although end-to-end models still struggled to outperform cascade systems, research in this area grew steadily because researchers believed that the potential of end-to-end models was higher and could address inherent issues in cascade systems, such as error propagation. After years of research and development, end-to-end models have now surpassed cascade systems in performance and are being deployed in industrial applications [2,3].
> >
> > Similarly, in our case, **while cascaded systems can achieve relatively good results currently, there are certain challenges that may only be addressed by end-to-end models in the future** (for example, achieving fluent speech generation under extremely low latency, and better perception of paralinguistic information in the input speech). Regarding the current issues with LLaMA-Omni, we believe that they can be addressed in the future. For example, increasing the data scale could improve speech generation quality. Therefore, **we believe that the short-term limitations of end-to-end models should not deter further exploration in this area.** We believe that our work is a pioneering effort in the field of end-to-end speech language models, and we hope it will inspire further research in this area in the future.
> >
> > ---
> >
> > **References:**
> >
> > [1] Berard et al., 2016. Listen and Translate: A Proof of Concept for End-to-End Speech-to-Text Translation.
> >
> > [2] Seamless Communication et al., 2023. Seamless: Multilingual Expressive and Streaming Speech Translation.
> >
> > [3] Huang et al., 2023. Speech Translation with Large Language Models: An Industrial Practice.

---

> > > ### Author Response · Authors · 2024-11-29
> > > **Further responses (2/2)**
> > >
> > > > Regarding the InstructS2S-200K
> > >
> > > **Regarding the statement "I believe this 'advantage' may diminish as the size of the text-LLM scales," we believe that this conclusion should not be drawn so easily.** Using synthetic data for training is currently one of the most common methods for fine-tuning LLMs. For example, in the post-training of LLaMA 3.1, the majority of the data was generated by LLaMA 2 and earlier versions of LLaMA 3 [4]. These synthetic data have proven to be effective for training models of similar (70B) or even larger (405B) scales. Therefore, we believe that data generated by a LLaMA-3-70B model can also be effectively used to align itself with speech interaction scenarios, although we currently lack the resources to verify this claim.
> > >
> > > Additionally, regarding your comment on prompt-based methods, as we understand it, this approach can only be applied to cascaded systems involving ASR + LLM + TTS. For end-to-end systems, enabling the model to understand and generate speech typically requires training the model. Whether training directly on existing instruction data or following our approach of instruction rewriting before training, catastrophic forgetting is an unavoidable issue during the training process. **In other words, catastrophic forgetting is not a result of our dataset construction method.**
> > >
> > > **In our view, comparing prompt-based methods with our approach isn't particularly meaningful. The former is the simplest method for aligning style in a cascaded system for speech interaction, while our approach is the most straightforward and effective method for building an end-to-end system.** We acknowledge that catastrophic forgetting in speech language models is an important issue, but it falls beyond the scope of this work.
> > >
> > > ---
> > >
> > > > This paper is more engineering and does not bring refreshing new insights
> > >
> > > **We disagree with the statement "this paper is more engineering and does not bring refreshing new insights."** We have already clarified the novelty of our work in our initial response and will not repeat it here. To the best of our knowledge, prior to our work, we have not seen any related research or open-source projects that built end-to-end speech language models based on open-source LLMs, capable of enabling high-latency and low-quality speech interaction. **Moreover, based on the feedback from other reviewers, we also believe our work contributes to the research community, rather than being merely an engineering effort.**
> > >
> > > ---
> > >
> > > >Redo almost all the experiment and analysis during the review stage
> > >
> > > **We would like to clarify that, although we have updated many of the experimental results, the overall content of our experiments remains unchanged.** In the initial version of our paper, we have provided results for LLaMA-Omni in both offline and streaming scenarios, including metrics such as ChatGPT Score, ASR-WER, and UTMOS at different latencies. During the review phase, we took the reviewers' feedback into account and added results of the cascaded baseline systems in streaming scenarios, along with corresponding improvements to the metrics.
> > >
> > > We believe that the extended discussion period and the option to upload revised PDFs for ICLR are designed to allow authors the time to refine their experiments and paper based on reviewer feedback more carefully. **Therefore, we believe that our work fully complies with ICLR's guidelines.**
> > >
> > > ---
> > >
> > > **In conclusion, despite any differences in opinion, we remain grateful for the effort you put into the review process. We understand that providing multiple rounds of detailed feedback is a rare and valuable contribution. Our responses are intended to facilitate a deeper discussion of differing viewpoints. Regardless of whether you decide to maintain or revise your score, we fully respect your decision. Once again, thank you for your time.**
> > >
> > > Best regards,
> > >
> > > Authors
> > >
> > > ---
> > >
> > > **References:**
> > >
> > > [4] Grattafiori et al., 2024. The Llama 3 Herd of Models.

---

> ### Comment · Reviewer_ppgU · 2024-11-29
>
> Thanks to the authors for their quick response.
>
> I think that 'prompting only works for cascaded system' is a good point, which I have missed. And I've listened the demo. Llama-Omni does sound more natural. I'll raise my score accordingly.
>
> Regarding the comparison between Cascade and Llama-Omni, I find one thing confusing—the ASR-WER for Llama-Omni (text) + Orca is highest when $\Theta=5$. The same problem also happens in SALMONN + Orca. The high ASR-WER leads to lower performance and may narrow the gap between Cascade and Llama-Omni. Usually, one would expect a streaming TTS to perform better when the chunk size is larger. Is there any explaination on this phenomenon?

---

> ### Author Response · Authors · 2024-11-30
> **Thank you for your quick reply!**
>
> Thank you for your quick reply!  We are glad that our response has addressed some of your concerns. Regarding why the ASR-WER is higher when $\Theta = 5$ compared to when $\Theta$ is smaller, especially when $\Theta = 1$, we have also noticed this issue. After reviewing some examples, we believe the reason is that **when $\Theta = 1$, the cascaded system is essentially "reading word by word", which may reduce the difficulty for ASR and result in a lower WER**. In contrast, with $\Theta = 5$, there are instances where incoherence or errors occur between words. Some cases are as follows:
>
> >**Case 1:**
> >
> >**Text:** The Northern Lights are caused by charged particles from the sun interacting with the Earth's magnetic field and atmosphere.
> >
> >$\Theta=1$: The northern lights are **caused by** charged particles from **the sun** interacting with the Earth's magnetic field and atmosphere.
> >
> >$\Theta=5$: The northern lights are **Cosby** charged particles from **Thiessen** interacting with the Earth's magnetic field and atmosphere.
> >
> >$\Theta=9$: The northern lights are **caused by** charged particles from **Sun** interacting with the Earth's magnetic field and atmosphere.
>
> >
> >**Case 2:**
> >
> >**Text:** Yes, human blood is typically a deep red color, but it can appear more purple or blue depending on the oxygen levels and the individual's health.
> >
> >$\Theta=1$: Yes, human blood is **typically a** deep red color, **but it** can appear more purple or blue depending on the oxygen levels and the **individual's health**.
> >
> >$\Theta=5$: Yes, human blood is **typicalia** deep red color. **Budit** can appear more purple or blue depending on the oxygen levels and the **individual shelf**.
> >
> >$\Theta=9$: Yes, human blood is **typically a** deep red color, **but it** can appear more purple or blue depending on the oxygen levels and the individual's health.
>
> An interesting observation is that when $\Theta = 5$, many errors occur at the 5th and 10th words, which coincides with the word chunk boundaries. **This suggests that most of the errors in the streaming TTS system are caused by the incoherence between word chunks.** When we further increased the word chunk size to $\Theta = 9$, we found that many of the errors were resolved, which aligns with the intuition that larger word chunks lead to better performance.
>
> We also investigated why Qwen2-Audio+Orca did not exhibit similar behavior at $\Theta = 1$. We found that for outputs consisting solely of English words, $\Theta = 1$ can achieve a lower WER than $\Theta = 5$, similar to what we observed with LLaMA-Omni and SALMONN. However, for outputs containing special characters, such as newline characters or Chinese characters, when these are split into individual chunks, the model often generates strange outputs, leading to higher WER. This may explain why Qwen2-Audio+Orca performs worse at $\Theta = 1$ compared to $\Theta = 5$ overall. This also highlights a potential advantage of end-to-end systems over cascade systems: there is no need to process intermediate text outputs to bridge the gap between LLM outputs and TTS inputs.
>
> We hope our response can address your concerns. Once again, thank you for your efforts in helping improve our work.

---

> ### Comment · Reviewer_ppgU · 2024-11-30
>
> Thank you for the explaination.
>
> I agree the weird behavior of Cascade model could be caused by the 'unnatural rhythm' of Orca, as the provided cases shows wrong ASR results generally appears at the word chunk boundaries (every 5 words). I believe this help address my concern and demonstrates the advantage of End-to-end approaches in audio generation.
>
> I'll raise my score to 6. Good luck!

---

> > ### Author Response · Authors · 2024-11-30
> > **Thank you for raising the score!**
> >
> > Thank you for your quick reply and for raising the score! I really appreciate the effort you've put into the review process. Wishing you a great day!

---

### Official Review · Reviewer_MWRj · 2024-11-02

**Soundness:** 2
**Presentation:** 3
**Contribution:** 3
**Rating:** 6
**Confidence:** 5

**Summary:**

This paper proposed a Speech-LLM model that can do speech-to-speech conversation with good latency and quality. The central design enables the low latency feature is the proposed streaming speech decoder.

**Strengths:**

This is probably the first public work that connects a speech-to-text Speech-LLM with a streaming text-to-speech model so as to enable low latency speech outputs.

**Weaknesses:**

1. The speech-to-text component of this model is not new (relevant works like SALMONN and Qwen2-audio) and the main addition is the text-to-speech component. The latter has a clear connection with the streaming TTS model research but the paper does not spend enough content to acknowledge and compare to this line of research. To elaborate on this, the paper should compare the proposed method with prior research in this space and motivate this unique CTC based design. The paper should also replace the baseline of SALMONN+TTS with SALMONN+streaming TTS so as to have a fair comparison and show the value of the proposed component.
2. The "response latency" used in this paper is misleading. It only considers the latency from streaming speech decoder, but in fact the non-streaming speech-to-text Speech-LLM also has latency which should be considered as "response latency". The paper fails to point this out and didn't give the overview of the latency breakdown given the proposed design
3. The audio quality of the proposed method is not good from the MOS score and is not compared with any external works, which challenges the value of this proposed CTC based design, especially considering there have been prior works in the streaming TTS area. It's hard to understand whether the proposed method is better or worse.
4. "On the other hand, the ASR model we use, Whisper-large-v3, has strong robustness. Even when the speech is somewhat discontinuous with smaller Ω," If this ASR model cannot identify the discontinuity and relevant quality issue, it may not serve well as a way to evaluate the intelligence of the generated speech. This is a limitation.

**Questions:**

please resolve the concerns in Weaknesses

---

> ### Author Response · Authors · 2024-11-23
> **Author Response (1/n)**
>
> Thank you for providing your valuable and constructive feedback! Please find below the responses to each comment.
>
> ---
>
> **[Q1] Comparison with research related to streaming TTS.**
>
> Thank you for pointing this out. As you mentioned, our speech decoder performs the task of streaming TTS. Following your suggestion, we investigate research on streaming TTS (also known as incremental TTS) and find that most incremental TTS approaches adopt model architectures similar to offline TTS models (e.g., Tacotron 2 [1]), and employ a **fixed lookahead strategy**, such as waiting for a few future words [2-3] or using a language model to predict several upcoming words [4-6]. A few other works have also explored using reinforcement learning [7] or Transducer architectures [8] to implement incremental TTS. Compared to these works, we believe our approach has the following advantages:
>
> 1. Previous incremental TTS models typically adopt the traditional encoder-decoder architecture, whereas our model employs a **decoder-only Transformer** structure, unified with the LLM. Based on the success of such architectures in language modeling, we believe this design is **more suitable for scaling up**.
> 2. Unlike the fixed lookahead strategies used in most prior works, we leverage the blank symbol from CTC to represent the "WAIT" action. This enables our model to adopt a more **flexible and adaptive strategy** for determining the timing of outputs.
> 3. Compared to methods based on reinforcement learning and Transducer architectures, our model is easier and more stable to train. Reinforcement learning approaches often face challenges with unstable training, while Transducer models require significant GPU memory due to the computational complexity of the loss function. In contrast, **CTC-based training is more memory-efficient and stable**.
> 4. Many previous incremental TTS works assume that the input has already been converted into phonemes. However, as pointed out by [9], a practical system also requires a frontend module to streamingly convert text into phonemes. Our work integrates all these steps into a unified end-to-end model, **avoiding error propagation** introduced by intermediate steps.
>
> **We have added discussions on streaming TTS works in the Related Work section of the revised PDF**. Additionally, we have replaced the TTS model in the baseline system with a streaming TTS model. The following part will present the corresponding experimental results.
>
> ---
>
> **[Q2] Comparison with cascaded baseline systems with streaming TTS.**
>
> Following your suggestion, we replace the TTS component of cascaded baseline systems with an **industrial TTS model, [Orca](https://github.com/Picovoice/orca)**, which **supports both streaming and offline speech synthesis**. When using Orca for streaming speech synthesis, we need to set a **word chunk size** $\Theta$, which means that speech synthesis is triggered every time $\Theta$ new words arrive. In our experiments, we varies $\Theta$ within the range of $[1, 3, 5, 7, 9]$ to control the response latency of cascaded systems.
>
> **We conducted a comprehensive re-evaluation of all models in both offline and streaming scenarios**. Specifically, we evaluate the model using the following metrics:
>
> 1. **ChatGPT Score [Updated]**: We use GPT-4o to rate the model's responses. Following the suggestions from reviewers ppgU and DXz7, we have revised the evaluation prompt (details in Appendix A of the revised PDF) to remove potential prior biases.
> 2. **ASR-WER [Unchanged]**: This metric evaluates the alignment between text and speech responses.
> 3. **UTMOS [Unchanged]**: This metric assesses the quality of the generated speech.
> 4. **WPS (Words Per Second) [New]**: This metric measures the speaking rate of the generated speech, which is calculated as the average number of words per second in the generated speech.
> 5. **Latency [Updated]**: Latency refers to the **total delay** between the user's speech input and the moment they hear the speech response. This can be further divided into the **decoding latency of the LLM** and **the latency introduced by the Vocoder or TTS model**.

---

> ### Author Response · Authors · 2024-11-23
> **Author Response (2/n)**
>
> The numerical results under different latency conditions for the cascade systems (SALMONN + Orca, Qwen2-Audio + Orca) and LLaMA-Omni are as follows:**[SALMONN + Orca]**
>
> | $\mathbf{\Theta}$ | Latency-LLM (ms) | Latency-TTS (ms) | Latency-Total (ms) | ChatGPT Score | ASR-WER | UTMOS   | WPS  |
> |--------------------|----------|----------|------------|---------------|---------|---------|------|
> | 1                 | 212.45   | 19.71    | 232.16     | 3.28          | 6.64    | 3.0947  | 1.86 |
> | 3                 | 316.59   | 35.08    | 351.67     | 3.09          | 8.65    | 3.7338  | 2.91 |
> | 5                 | 428.79   | 32.08    | 460.87     | 3.04          | 9.23    | 3.7750  | 3.18 |
> | 7                 | 536.47   | 45.90    | 582.37     | 3.06          | 8.49    | 3.7972  | 3.20 |
> | 9                 | 659.57   | 69.38    | 728.95     | 3.23          | 7.07    | 3.8060  | 3.22 |
> | Offline           | 4274.48  | 1049.59  | 5324.07    | 3.40          | 3.78    | 3.8286  | 3.31 |
>
> **[Qwen2-Audio + Orca]**
>
> | $\mathbf{\Theta}$ | Latency-LLM (ms) | Latency-TTS (ms) | Latency-Total (ms) | ChatGPT Score | ASR-WER | UTMOS   | WPS  |
> |--------------------|----------|----------|------------|---------------|---------|---------|------|
> | 1                 | 289.46   | 19.15    | 308.61     | 2.79          | 25.25   | 2.8597  | 1.90 |
> | 3                 | 381.66   | 38.19    | 419.85     | 2.95          | 13.30   | 3.5529  | 2.91 |
> | 5                 | 470.55   | 38.64    | 509.19     | 2.93          | 13.00   | 3.5865  | 3.08 |
> | 7                 | 568.22   | 52.95    | 621.17     | 3.07          | 10.59   | 3.5739  | 3.11 |
> | 9                 | 675.55   | 81.02    | 756.57     | 3.14          | 9.81    | 3.6016  | 3.13 |
> | Offline           | 7062.93  | 2361.49  | 9424.42    | 3.38          | 6.77    | 3.6119  | 3.30 |
>
> **[LLaMA-Omni]**
>
> | $\mathbf{\Omega}$ | Latency-LLM (ms) | Latency-Vocoder (ms) | Latency-Total (ms) | ChatGPT Score | ASR-WER | UTMOS   | WPS  |
> |--------------------|----------|--------------|------------|---------------|---------|---------|------|
> | 10                | 206.03   | 30.15        | 236.18     | 3.54          | 9.84    | 3.2304  | 2.76 |
> | 20                | 236.18   | 45.23        | 281.41     | 3.56          | 9.91    | 3.4748  | 2.75 |
> | 40                | 301.51   | 45.23        | 346.73     | 3.52          | 10.37   | 3.6688  | 2.74 |
> | 60                | 361.81   | 50.25        | 412.06     | 3.52          | 10.47   | 3.7549  | 2.74 |
> | 80                | 432.16   | 55.28        | 487.44     | 3.50          | 10.70   | 3.7858  | 2.73 |
> | 100               | 497.49   | 65.33        | 562.81     | 3.49          | 10.71   | 3.8242  | 2.74 |
> | Offline           | 1542.71  | 211.06       | 1753.77    | 3.47          | 10.82   | 3.9296  | 2.73 |
>
> >  **Note**: The latency of LLaMA-Omni has slightly changed compared to the submitted version due to variations in machine performance. To ensure fairness, we reevaluated the latency of all systems under the same conditions.
>
> From the experimental results, we observe the following:
>
> 1. Both the cascade systems and LLaMA-Omni **can achieve latencies below 320ms** (the average audio latency of GPT-4o) and allow latency control by adjusting hyperparameters. **Across all latency conditions, LLaMA-Omni consistently achieves higher ChatGPT Scores compared to the cascade baseline systems**.
>
> 2. In the offline scenario, the cascade systems achieve lower ASR-WER than LLaMA-Omni, demonstrating the superior performance of industrial TTS models. However, **in the streaming scenario, as latency decreases, we observe a significant performance drop in the cascade systems compared to the offline scenario** (evidenced by a decrease in ChatGPT Score and an increase in ASR-WER). In contrast, **LLaMA-Omni shows minimal changes in ChatGPT Score and ASR-WER as latency varies**, and even exhibits performance improvements in some cases.
> 3. In terms of speech quality, all systems experience a decline in quality as latency decreases, primarily due to an increase in discontinuities within the speech. LLaMA-Omni achieves speech quality comparable to that of SALMONN+Orca.
> 4. From the WPS metric, **LLaMA-Omni's speech rate remains almost unaffected by latency changes**. In contrast, **the overall speech rate of the cascade systems decreases significantly under low-latency conditions**, primarily due to increased pauses between word chunks, which significantly affects the overall rhythm and prosody of the speech. For example, at the lowest latency ($\Theta=1$), the overall speech rate drops to less than 60% of the offline scenario. This indicates that cascade systems face significant challenges under extremely low-latency conditions.

---

> ### Author Response · Authors · 2024-11-23
> **Author Response (3/3)**
>
> For the above results, **we have included line charts in the revised PDF illustrating the variation of each metric across different latency conditions for all models**. These charts provide a more intuitive comparison of the performance of different models. Please refer to the updated version of the paper. **We have also provided audio samples at** https://llama-omni.github.io/, allowing for a more intuitive understanding of the speech generated by different models under various latency conditions through listening.
>
> We also conducted **human evaluations** to investigate human preferences in real-time speech interaction scenarios. Specifically, we performed **side-by-side comparisons** between LLaMA-Omni ($\Omega=40$, latency=347ms) and two cascade systems: SALMONN+Orca ($\Theta=3$, latency=352ms) and Qwen2-Audio+Orca ($\Theta=3$, latency=420ms). The comparisons evaluated the systems in terms of the **helpfulness** of the responses and the **naturalness** of the generated speech, using a win/tie/lose framework. The evaluation results are as follows:
>
> | Helpfulness                     | Win  | Tie  | Lose |
> | ------------------------------- | ---- | ---- | ---- |
> | LLaMA-Omni vs. SALMONN+Orca     | 48   | 28   | 24   |
> | LLaMA-Omni vs. Qwen2-Audio+Orca | 50   | 20   | 30   |
>
> | Naturalness                     | Win  | Tie  | Lose |
> | ------------------------------- | ---- | ---- | ---- |
> | LLaMA-Omni vs. SALMONN+Orca     | 44   | 33   | 23   |
> | LLaMA-Omni vs. Qwen2-Audio+Orca | 42   | 20   | 38   |
>
> From the results, we observed that LLaMA-Omni achieved higher win rates in both helpfulness and naturalness, demonstrating that **LLaMA-Omni's responses better align with human preferences**. More details about the human evaluation can be found in Section 4.6 of the revised PDF.
>
> ---
>
> **[Q3] The "response latency" used in this paper is misleading.**
>
> Apologies for the misunderstanding. In our paper, **latency does not only refer to the streaming speech decoder's delay** but comprehensively accounts for the total delay from when the user finishes their speech input to when they hear the speech response. This includes the decoding latency of LLaMA-Omni (covering both the LLM's text generation and the speech decoder's speech generation) as well as the latency of the vocoder synthesizing the speech. **We have clarified this in the table above and in the revised paper, where we explicitly list the latency for each step**.
>
> ---
>
> **[Q4] The audio quality of the proposed method is not good from the MOS score and is not compared with any external works.**
>
> We have added the UTMOS scores for the cascade systems in above tables, whose speech is generated by an industrial TTS model. The results show that LLaMA-Omni achieves UTMOS scores comparable to or slightly higher than SALMONN + Orca across different latency conditions, indicating that the speech quality generated by LLaMA-Omni is satisfactory.
>
> ---
>
> **[Q5] ASR-WER may not serve well as a way to evaluate the intelligence of the generated speech.**
>
> Thank you for your suggestion. ASR-WER is a commonly used metric in TTS that evaluates whether the **content** of the generated speech aligns with the corresponding text. While speech discontinuities may not be well reflected in the ASR-WER metric, they can be perceived through other metrics such as UTMOS and human evaluations. **We believe these metrics together provide a comprehensive evaluation of the generated speech from multiple dimensions, with ASR-WER being one of the most reliable methods for assessing speech content accuracy.**
>
> ---
>
> **References:**
>
> [1] Shen et al., 2017. Natural TTS Synthesis by Conditioning WaveNet on Mel Spectrogram Predictions.
>
> [2] Ma et al., 2020. Incremental text-to-speech synthesis with prefix-to-prefix framework.
>
> [3] Stephenson et al., 2020. What the future brings: Investigating the impact of lookahead for incremental neural tts.
>
> [4] Saeki et al., 2021. Incremental text-to-speech synthesis using pseudo lookahead with large pretrained language model.
>
> [5] Saeki et al., 2021. Low-latency incremental text-to-speech synthesis with distilled context prediction network.
>
> [6] Liu et al., 2022. From start to finish: Latency reduction strategies for incremental speech synthesis in simultaneous speech-to-speech translation.
>
> [7] Mohan et al., 2020. Incremental Text to Speech for Neural Sequence-to-Sequence Models using Reinforcement Learning.
>
> [8] Chen et al., 2021. Speech-T: Transducer for Text to Speech and Beyond.
>
> [9] Dekel et al., 2023. Speak While You Think: Streaming Speech Synthesis During Text Generation.

---

> > ### Author Response · Authors · 2024-11-25
> > **Looking forward to your feedback!**
> >
> > Dear Reviewer MWRj,
> >
> > Thank you once again for your valuable feedback. We have conducted additional experiments and made revisions to the paper based on your suggestions. As the discussion phase is nearing its conclusion, we would like to know if our responses have addressed your concerns. We look forward to hearing from you.
> >
> > Best,
> > Authors

---

> ### Author Response · Authors · 2024-11-26
> **Thank you for raising the score!**
>
> Dear Reviewer MWRj,
>
> Thank you for raising the score! Once again, we sincerely appreciate the valuable feedback you provided, which has been incredibly helpful in improving our work.
>
> Best,
>
> Authors

---

### Author Response · Authors · 2024-11-23
**Global Response**

Dear AC and reviewers,

Thank you for your constructive suggestions. We have provided detailed responses to each of your concerns. We sincerely appreciate your efforts in helping us improve our work. Taking your feedback into account, we have refined our evaluation metrics and conducted a comprehensive re-evaluation. **The revised version of our paper has been uploaded, and the core modifications are summarized as follows:**

- We divided the experiments into **offline** and **streaming** scenarios and conducted evaluations for all models in both settings.
- We replaced the TTS model in the cascading system with an **industrial TTS model that supports both offline and streaming synthesis**, enabling a more comprehensive comparison across different latency conditions.
- We revised the ChatGPT scoring prompts to remove any potential bias.
- We corrected an error that was inadvertently introduced in the calculation of the ASR-WER metric.
- We introduced a new metric, Words Per Second (WPS), to evaluate the speech rate. We  also provide more specific information on latency, detailing the latency at each stage.
- We conducted **human evaluations** to analyze human preferences for different models.
- We provided **audio samples** at https://llama-omni.github.io/ to offer a more intuitive understanding of the speech responses generated by different models.
- We added discussions on streaming TTS works in the related work.

---

### Author Response · Authors · 2024-11-30
**Thank you for your efforts in the review process!**

Dear AC and Reviewers,

We would like to express our sincere gratitude once again for the effort you have put into the review process. We are pleased to have been able to address all the reviewers' concerns during the discussion phase. The insightful feedback provided has significantly helped us to improve our work. We truly appreciate the time and attention the reviewers have dedicated to engaging in this process, offering timely and detailed feedback, and generously increasing the scores.

We are delighted to have received all positive scores (8, 6, 6, 6) in the end, and we hope that our work will make a meaningful contribution to the community.

Thank you once again to the AC and all the reviewers for your valuable efforts!

Best regards,

Authors

---

### Meta-Review · Area_Chair_Y2FM · 2024-12-21

**Metareview:**

This paper proposed LLaMA-Omni, a model architecture designed for low-latency speech interaction with LLM which is a very important research area  with the representative works such as GPT-4o and Moshi. The proposed architecture integrates a pretrained speech encoder, an adaptor, an LLM, and a streaming speech decoder.  With such a structure, the model can simultaneously generates text and speech responses directly from speech instructions with the latency as low as 236ms. To align the model with speech interaction scenarios, the authors also constructed a dataset with 200K speech instructions and corresponding speech responses. Experiment results showed that LLaMA-Omni is superior than previous speech LLMs.

The paper presents extensive experiments in offline and streaming scenarios. The proposed work stands out due to its low-latency streaming processing, unlike most speech LLMs. In their rebuttal, the authors used an industrial TTS model to strengthen the baseline for cascaded systems, making the LLaMA-Omni's improvements more convincing. The initial submission has the issues of calculating ASR-WER metrics and using ChatGPT scoring which were pointed out by reviewers. These were all fixed during the rebuttal. Furthermore, the authors have conducted human evaluation to compare different models.

Overall, after revision, the paper is much stronger than the initial submission. Streaming speech-LLM with low latency is an area with very limited works at this moment. The proposed work is a good plus to the field.

In summary, this paper's strength lies in its focus on streaming speech LLM scenarios. The proposed method demonstrates superior performance compared to the cascaded system in low-latency situations, which is highly beneficial for spoken chat applications. However, a noted weakness is that the proposed method does not consistently outperform the cascaded system across all configurations, with occasional instances where the cascaded baseline exhibits an unusually high word error rate.

**Additional Comments On Reviewer Discussion:**

The initial scores for this paper were 3, 3, 5, and 6. The reviewers raised numerous questions. During the rebuttal process, the authors added a substantial number of experiments to demonstrate the effectiveness of the proposed methods. The paper has been significantly improved due to the authors’ extensive efforts. Consequently, the final scores are now 6, 6, 6, and 8, indicating a considerable improvement during the rebuttal process. While the original submission was not acceptable, the revised version meets the standards for publication. I concur with Reviewer ppgU's observation that submitting an incomplete manuscript and conducting most of the experiments and analysis during the review stage is not advisable.

---

### Decision · Program_Chairs · 2025-01-22

Accept (Poster)